# Integrated transcriptome study of the tumor microenvironment for treatment response prediction in male predominant hypopharyngeal carcinoma

Yang Zhang [1,4] ✉, Gan Liu [2,3,4] ✉, Minzhen Tao [2,4], Hui Ning[2], Wei Guo[1], Gaofei Yin[1], Wen Gao[1], Lifei Feng [1], Jin Gu [2], Zhen Xie [2] ✉ & Zhigang Huang [1] ✉

The efficacy of the first-line treatment for hypopharyngeal carcinoma (HPC), a predominantly male cancer, at advanced stage is only about 50% without reliable molecular indicators for its prognosis. In this study, HPC biopsy samples collected before and after the first-line treatment are classified into different groups according to treatment responses. We analyze the changes of HPC tumor microenvironment (TME) at the single-cell level in response to the treatment and identify three gene modules associated with advanced HPC prognosis. We estimate cell constitutions based on bulk RNA-seq of our HPC samples and build a binary classifier model based on non-malignant cell subtype abundance in TME, which can be used to accurately identify treatment-resistant advanced HPC patients in time and enlarge the possibility to preserve their laryngeal function. In summary, we provide a useful approach to identify gene modules and a classifier model as reliable indicators to predict treatment responses in HPC.

Head and neck cancer, one of the most common cancers worldwide, with nearly 870,000 new cases and 440,000 deaths occurring each year[1], ~90% of which are head and neck squamous cell carcinoma (HNSCC)[2], including cancers in the lip and oral cavity, larynx, naso-pharynx, oropharynx and hypopharynx[3]. Anatomically, the hypo-pharynx is commonly defined by its subsites, including the lateral pharynx, posterior pharyngeal wall, piriform sinuses, and the post-cricoid region leading to the esophageal inlet. Hypopharyngeal carci-noma (HPC) is mostly diagnosed in males, and the difference in late-life incidence between men and women is more than tenfold in East Asian[4]. In addition, due to its hidden sites and being asymptomatic at early stage, HPC is often diagnosed at advanced stage. Therefore, HPC is a relatively rare cancer and accounts for ~3% of all HNSCCs, but it has the worst prognosis among all HNSCCs with a 5-year overall survival rate at about 30–35%[5]. In addition, the 3-year and 5-year survival rates of advanced HPC are 22.86% and 11.43%, respectively, according to ret-rospective researches in Beijing Tongren Hospital[6], which is a large laryngeal cancer and HPC diagnosis and treatment center in North China. Since cetuximab that targets EGFR was approved for HNSCC in 2006, it combined with radiotherapy or chemotherapy has become the first-line therapy for the treatment of HPC[5,7]. In practice, we apply TPF (taxol, cisplatin, 5-FU) induction chemotherapy plus cetuximab,

[1]Department of Otolaryngology Head and Neck Surgery, Beijing Tongren Hospital, Capital Medical University, Key Laboratory of Otolaryngology Head and Neck Surgery (Capital Medical University), Ministry of Education, 100730 Beijing, China. [2]MOE Key Laboratory of Bioinformatics and Bioinformatics Division, Center for Synthetic and System Biology, Department of Automation, Beijing National Research Center for Information Science and Tech-nology, Tsinghua University, 100084 Beijing, China. [3]Tsinghua-Peking Joint Center for Life Sciences, Tsinghua University, 100084 Beijing, China. [4]These authors contributed equally: Yang Zhang, Gan Liu, Minzhen Tao. ✉e-mail: zhangyangent@163.com; wz_liugan@163.com; zhenxie@tsinghua.edu.cn; huangzhigang1963@163.com

the combined treatment, for advanced HPC patients to reduce tumor volume, and then re-evaluate the possibility of radical surgical resection of the HPC tumor. In our former retrospective study with 63 HPC patients, the objective response rate to the combined treatment is only 52%, including partial and complete decreased tumor mass[6]. The treatment-resistant patients' conditions are deteriorated, and these patients missed the best time for surgery or other possible therapies.

In the face of such a severe situation, apart from mutations of HNSCC pan-cancer genes such as TP53, CDKN2A, EGFR and the dysfunction of WNT pathway[8], no effective biomarker has been identified to infer HPC progression, prognosis and combined treatment response for HPV-negative patients. Pan-cancer analyses reveal that malignant tumor cells are highly heterogeneous, which drives neoplastic progression and therapeutic resistance[9]. Therefore, recent researches define gene modules and use the corresponding gene sets for characterizing tumor heterogeneity and predicting prognosis[10–12]. In addition, various subtypes of conserved non-malignant cells in tumor microenvironment (TME), such as immune cells, endothelial cells, and cancer-associated fibroblasts (CAFs) are related with tumor prognosis[13–16]. For instance, the increase of CD8 + T cells in tumor-infiltrating lymphocytes of HPC indicates a good prognosis[17]. However, current strategies for transcriptomic analyses of HPC are primarily based on bulk samples, and therefore these approaches lack the resolution and accuracy to discover effective and reliable biomarkers for risk estimation in both prognosis and clinical stratification. Recent advances in single-cell RNA sequencing (scRNA-seq) have been utilized for depicting heterogeneous malignant tumor cells and complex cell constitutions in TME for several cancer types, such as melanoma, lung adenocarcinoma, and nasopharyngeal carcinoma (NPC)[11,12,18–20].

In this study, we provide a comprehensive and unique resource revealing the landscape of HPC TME at single-cell resolution. Through systematic analyses based on scRNA-seq and bulk RNA-seq data from clinical samples of predominantly male participants, we uncover three functional gene modules of malignant tumor cells associated with prognosis, and establish the relationship between non-malignant subtypes' composition and patients' responses to the combined therapy, which offer important clinical implications and may help avoid treatment delays in practice in future.

## Results

### Clinical features and single-cell landscapes of collected HPC samples from different groups

In clinical practices, treatment-naive HPC patients diagnosed by pathological phenotypes and corresponding CT scanning, usually accepted TPF induction chemotherapy plus cetuximab. After 6–8 weeks of one combined treatment cycle, the patients received the second CT scanning to evaluate the curative effect and to determine the following therapeutic strategy. Based on responses of patients to the combined treatment, clinical collected samples were divided into four groups: called responder before treatment (RBT), responder after treatment (RAT), non-responder before treatment (NBT), and non-responder after treatment (NAT) groups (Fig. 1a). Because HPC is a relatively rare cancer, it consumed us several years to establish our own HPC cohort, in which 44 samples from 44 individual patients were collected and divided into above four groups (Supplementary Table 1) with transcriptomic quantification by bulk RNA-seq. Survival analysis showed that patients in the RBT group had significantly increased survival rates than those in the NBT group (Fig. 1b and Supplementary Fig. 1). In order to capture features of heterogeneous malignant tumor cells and complex TME in HPC for potential effective biomarkers identification, we collected additional samples for scRNA-seq and combined with the above cohort to do systematic bioinformatics analyses and related validations (Fig. 1c).

We generated scRNA-seq profiles from 12 advanced HPC tumor samples, 1 lymph metastasis sample, and 2 samples of non-malignant

tissues (NT), totally 15 clinical samples from 8 patients (Supplementary Tables 2 and 3). Fresh biopsies were rapidly digested into single-cell suspensions and quantitated by droplet-based 10x Genomics Chromium scRNA-seq platform. Overall, we captured the transcriptomes of 6 major cell types with 89,094 cells after qualifying control filters (see "Methods"; Supplementary Fig. 2 and Supplementary Table 3). According to the expression of canonical marker genes, 19456 EPCAM + epithelial cells, 6859 CLDN5 + endothelial cells, 9115 COL1A1 + fibroblast cells, 16012 CD79A + B cells, 12210 LYZ + myeloid cells, 25442 CD3E + T and NCAM1 + NK cells were identified (Fig. 1d, e and Supplementary Fig. 3a, b). More expressed genes were detected in epithelial cells, endothelial cells and fibroblast cells than those in immune cells (Supplementary Fig. 3c). Moreover, the majority cells in the groups of Lymph and NT were T cells and B cells, whereas various cell subtypes were enriched in tumor samples (Supplementary Fig. 3d). Multiplex immunohistochemistry (mIHC) results confirmed the existence of these cell types (Fig. 1f). With mIHC plots, we observed that there were more infiltrating immune cells in groups of RBT and RAT (30.16% and 29.34%, respectively), compared with those in NBT and NAT groups (0.42% and 2.65%, respectively). Moreover, most of the stained epithelial cells were identified as tumor cells according to the downstream scRNA-seq data analyses. The large patch of stained epithelial cells in RBT was split into small clumps after effective combined treatment in RAT, with no significant change in cell proportions (31.33% and 34.85% in RBT and RAT, respectively). However, stained epithelial cells were more abundant in groups of NBT and NAT (55.19% and 80.99%, respectively).

### Three functional gene modules in heterogeneous malignant tumor cells of HPC

Malignant tumor cells were distinguished by inferring large-scale chromosomal copy-number variations (CNVs) within epithelial cells in each tumor and lymph sample, in which epithelial cells from NT group were used as reference (Fig. 2a)[21]. Totally, 19,207 malignant tumor cells from 13 samples were identified. Clustering of malignant cells revealed sample-specific clusters, indicating a high degree of inter-tumoral heterogeneity (Fig. 2b and Supplementary Fig. 4a). Considering the small percentage of malignant cells in RAT and lymph groups, we excluded these cells for further analyses (Supplementary Fig. 4b). Then we grouped malignant tumor cells from groups of RBT, NBT, and NAT, which exhibited improved clustering (Fig. 2c). Although the featured genes of each sample were different, they had similar expression pattern in biological function modules (Fig. 2d). For example, immune-related genes such as HLA-A, BCL6, S100P showed relatively high expressed in RBT group, whereas expression levels of genes associated with stemness and drug-resistance were increased in NBT and NAT groups. Notably, with the lowest expression level of tumor-suppressive genes in NBT, the representative gene CDKN2A was related with poor prognosis in both our HPC cohort and public NPC cohort[22] (Supplementary Fig. 4c). Pathway analyses revealed further differences among malignant cells in these three groups. Cells from the RBT group were highly activated in epidermal cell differentiation pathways, while those in the NBT group were enriched in Ribosome and DNA replication pathways (Supplementary Fig. 4d). As for malignant cells in NAT, several immunological pathways were reactivated after drug treatment (Supplementary Fig. 4d). These results suggested that functional gene modules, rather than individual genes, were more appropriate for depicting tumor transcriptional variability.

We applied non-negative matrix factorization (NMF) to malignant tumor cells from RBT, NBT and NAT groups for deciphering underlying gene modules with R package NMF[23]. With suitable preprocessing and rank selections (Supplementary Fig. 5a), we extracted 5 functional gene modules through hierarchical clustering, including Epi_development, EMT_extended, Cell-cycle, Immunity and Ribosome modules (Fig. 2e and Supplementary Table 4). Compared to RBT group, the

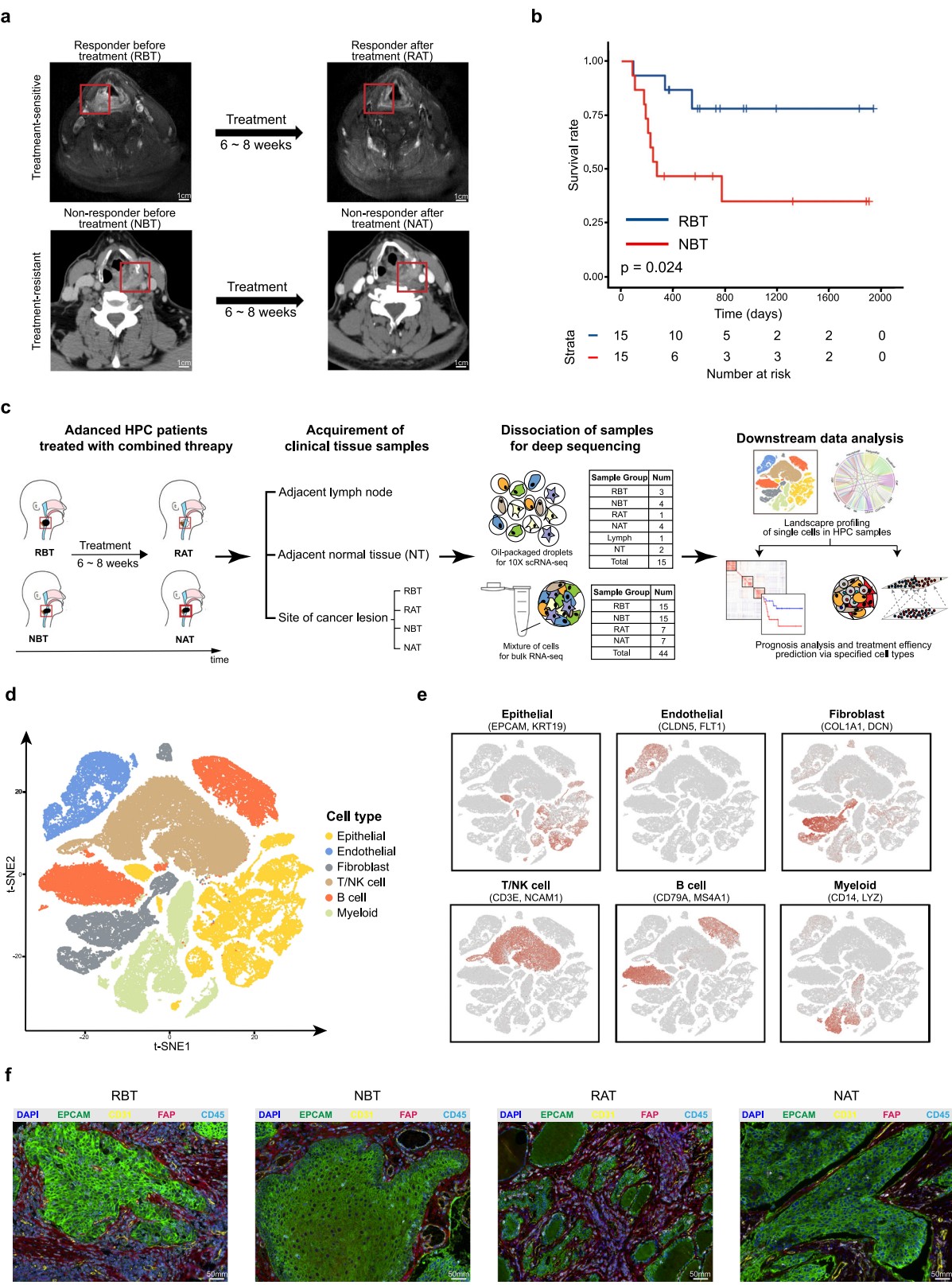

expression score of the Ribosome module was relatively increased in the NBT group and decreased in NAT group (Supplementary Fig. 5b, c). However, the transcriptional pattern of Immunity module showed in a reverse way. Samples in NBT groups had significantly low scores (Supplementary Fig. 5d, e), suggesting that a stronger interaction between immune cells and tumor cells may serve as a potential indicator for effective response to combined therapy. For the other three

functional gene modules, they exhibited the same trends in both scRNA-seq and bulk RNA-seq expression datasets separately (Fig. 2f–k). Specifically, the scores of EMT_extended and Cell-cycle modules showed an increased trend in NBT group than that in RBT group (Fig. 2f, i, g, j and Supplementary Fig. 5f, g), whereas the activity of the Epi_development module was significantly lower than that in RBT group (Fig. 2h, k and Supplementary Fig. 5h). Furthermore,

**Fig. 1 | Clinical features and single-cell landscapes of collected HPC samples from different groups. a** Radiological features of patients with different responses to one cycle of the combined treatment enrolled in the study. Red boxes indicate the tumor lesions, and tags show the names of groups. **b** Kaplan–Meier plot of survival analysis for patients in RBT and NBT groups. *P* value was calculated by the log-rank test. **c** An experimental scheme diagram highlighting the overall study design and downstream analyses. **d** t-SNE plot of overall 89094 single cells grouped into six major cell types. Each dot represents one single cell, colored according to cell type. **e** Normalized expressions of canonical marker genes for each cell type. The depth of color from gray to red represents low to high expression. **f** Representative images of multiplex immunohistochemistry (mIHC) for HPC tumor samples from RBT, NBT, RAT, and NAT groups. Samples were performed with anti-EPCAM, anti-CD31, anti-FAP, and anti-CD45 antibodies for epithelial cells, endothelial cells, fibroblast cells and immune cells identification separately. DAPI was used as a nuclear counterstain. Images are representative of three biological replicates.

survival analyses revealed that three gene modules could be used for prognosis prediction for both HPC (Fig. 2l–n and Supplementary Fig. 6a–c) and NPC cohorts (Supplementary Fig. 6d), in which patients with higher expression of EMT_extended and Cell-cycle modules, lower expression of Epi_development module were associated with poor prognosis. The above results indicate that these three functional gene modules have the potential to serve as predictive biomarkers in clinical.

## T-cell clustering and state analysis in HPC

Immune cells are composed of three major types, namely B cell, T/NK cell, and myeloid cell. Subclustering analyses identified distinct subtypes (Fig. 3a and Supplementary Fig. 10a). In the subset of T and NK cells, apart from the traditional four cell types (Supplementary Fig. 7a–c), we further categorized them into 14 subtypes based on scRNA-seq profiles, including 7 subtypes of CD8 + T cells (C1_CD8_naive, C2_CD8_memory, C3_CD8_cytotoxic1, C4_ CD8_cytotoxic2, C5_ CD8_cytotoxic3, C6_ CD8_exhaust and C7_ CD8_Ebo), 6 subtypes of CD4 + T cells (C8_Treg_naive, C9_Treg_act, C10_Treg_Ebo, C11_Th_naive, C12_Th_act, and C13_Th_exhaust), and 1 NK cell subtype (C14_NK) (Fig. 3a).

For CD8 + T cells, naive (CD8_C1; TCF7, LEF1), Ebo (CD8_C7; MKI67, NUSAP1)[24], cytotoxic (C3_CD8, C4_CD8, C5_CD8; GZMA, GZMK) and exhausted subtypes (CD8_C6; PDCD1, TIGIT) were identified according to expressions of marker genes (Fig. 3b and Supplementary Fig. 7d)[11,20]. The inferred developmental trajectory of CD8 + T cells exhibited a branched structure, with C1_CD8_naive as the root, C3_CD8_cytotoxic1, C4_CD8_cytotoxic2, C5_CD8_cytotoxic3 as intermediate states and two kinds of exhausted subtypes as the ends (Supplementary Fig. 7e). In addition, we scored the expression levels of genes related to corresponding functional pathways in four closely related CD8 + T subtypes (Supplementary Table 4), in which gradually increased cytotoxic scores from C3_CD8_cytotoxic1, C4_CD8_cytotoxic2, C5_CD8_cytotoxic3 to C6_CD8_exhaust, as well as high exhausted scores in C5_CD8_cytotoxic3 and C6_CD8_exhaust were revealed (Fig. 3c). These observations suggested that C4_CD8_cytotoxic2 should function as the main cytotoxic subtype and C5_CD8_cytotoxic3 was in an intermediate transition state towards exhaustion. Regulon[25,26] and GSVA pathway analyses also supported the above inference (Fig. 3d and Supplementary Fig. 7f). To be specific, C4_CD8_cytotoxic2 exhibited high expression levels in effectors such as EOMES and STAT1/2[27,28], while pathways of lymphocyte activation and negative regulation of immune were enriched in C6_CD8_exhaust. When comparing the proportions of subtypes among groups, we found that C1_CD8_naive was the majority in the lymph group and C3_CD8_cytotoxic1 accounted for the largest proportion in NT and RAT (Supplementary Fig. 7g). For samples from RBT, NBT and NAT, higher percentages of C4_CD8_cytotoxic2 and C6_CD8_exhaust were observed in RBT and NAT groups respectively (Fig. 3e and Supplementary Fig. 7g). In addition, cells with higher cytotoxic score and lower exhausted score were most abundant in RBT among the three groups (Q4: 53.66% vs 36.93%, and 41.97%), indicating a more activate state of RBT group (Fig. 3f).

For CD4 + T cell, we identified three Treg cell subtypes (C8_Treg_naive, C9_Treg_act, C10_Treg_Ebo; FOXP3, IL2RA) and three Th subtypes (C11_Th_naive, C12_Th_act, C13_Th_exhaust) (Fig. 3a and Supplementary Fig. 8). C8_Treg_naive had higher naive scores, while

C9_Treg_activity showed higher expression levels of TregStability and chemokines than that of C10_Treg_Ebo (Fig. 3g and Supplementary Table 4). When comparing the proportions of CD4 + Treg subtypes, we found that RBT group possessed fewer cells with high feature scores of TregStabillity and chemokines (Q1:24.08% vs 35.46%) in contrast to NBT group, suggesting a less immune-suppressive state of RBT group (Fig. 3h).

## Myeloid and B-cell clustering and state analysis in HPC

We analyzed the single-cell transcriptomes of 12,210 myeloid cells and then grouped them into 9 subtypes based on the expression of canonical markers, including Myeloid_C1_neutrophil, Myeloid_C2_mast, Myeloid_C3_monocyte, Myeloid_C4_pDC, Myeloid_C5_moDC, Myeloid_C6_cDC1, Myeloid_C7_cDC2, Myeloid_C8_macrophageM1 and Myeloid_C9_macrophageM2 (Fig. 3a and Supplementary Fig. 9a)[20,29,30]. For the subset of myeloid cells, the inferred developmental trajectory exhibited a branched structure, with monocyte as the root and two kinds of macrophage subtypes as the ends separately (Supplementary Fig. 9b). Calculating the M1 and M2 signature scores based on reported marker genes[31], we found M1-like pro-inflammatory signature scores were almost similar between macrophageM1 and macrophageM2 subtypes, while macrophageM2 had obviously higher immune-suppressive signature scores (Fig. 3i). M2-like marker genes, such as CD163[32], MSR1[33], and angiogenesis marker genes, like MMP9, VEGFA[29], were specifically drawn for presentation by violin plots (Fig. 3j). When coming the comparison of macrophage proportions among groups, we observed that the percentage of macrophage cells with both higher M1 and M2 signature scores was higher in NBT group than that of RBT group (Q1: 53.05% vs 37.84%) (Fig. 3k), and there existed more macrophageM2 in NBT and NAT compared to those in RBT and RAT groups (Supplementary Fig. 9c). As for four subtypes of DC cells, gene set enrichment analysis and expression level estimation of functional gene sets showed that pDC was a GZMB-mediated killer subset[34] and cDC2 tended to be a well-matured immunosuppressive DC subset with high expression of LAMP3, while moDC and cDC1 were DC subsets mainly responsible for antigen presentation and they had a relatively higher cell proportions in RBT group (Supplementary Fig. 9a, d–f)[20].

16,012 B cells were detected and annotated into 5 cell subtypes, including Bcell_C1_GC, Bcell_C2_MemoryInter, Bcell_C3_memory, Bcell_C4_plasama_IgA, and Bcell_C5_plasma_IgG (Supplementary Fig. 10a, b). Bcell_C1_GC cells were germinal center cells with high enrichment of pathways related to cell division and DNA replication, Bcell_C2_MemoInter and Bcell_C3_memory cells functioned as immunological memory cells with specific high expressions of CD19, LTB and IGHD, and Bcell_C4_plasma_IgA and Bcell_ C5_plasma_IgG cells played an immunological killing effect with specific high expressions of XBP1, IGHG and IGHA (Supplementary Fig. 10b, c). A rational developmental trajectory was depicted for these B cells (Supplementary Fig. 10d). Moreover, Bcell_C1_GC, Bcell_C2_MemoryInter were abundant in lymph and NT groups as expectations, while various types of B cells existed in NBT group, indicating B cells might play a role in TME of NBT group (Supplementary Fig. 10e).

## Two featured endothelial subtypes identified in HPC

Apart from infiltrating immune cells, the TME is also composed of a complex milieu of cell types including CAFs which make up the tumor

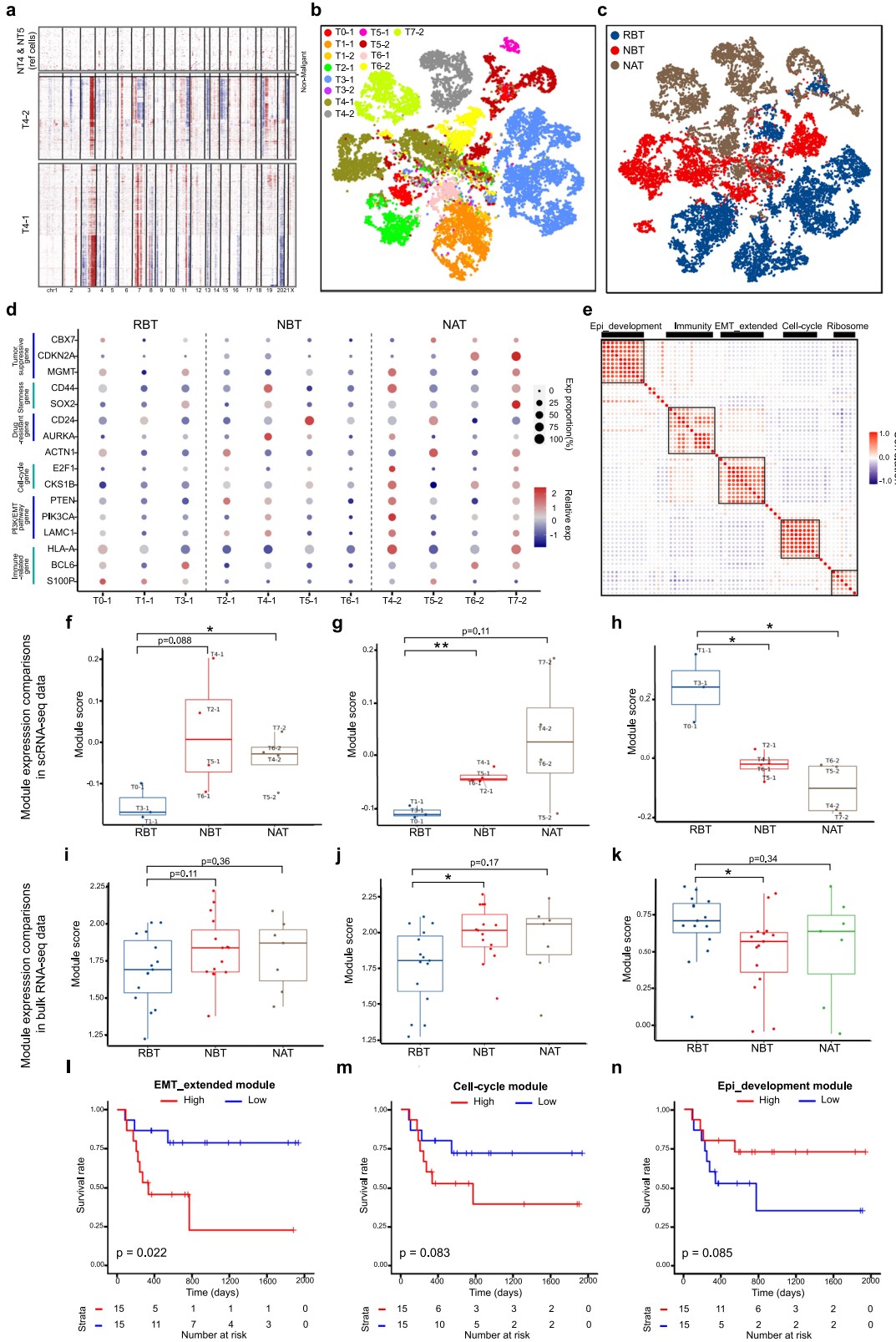

stroma, and endothelial cells (ECs) which line the lumens of blood and lymphatic vessels. Recent advances reveal that vascular endothelial cells are heterogenous and can function in different ways[35–37]. Some prioritize to sprout and migrate from a blood vessel (so-called tip-like cells), and others are relatively more static (so-called stalk-like cells)[35]. For endothelial cells, we identified one lymphoid endothelial subtype (Endo_C1_EndoLym) with high expression of PDPN, PROX1, LYVE1, and

two vascular endothelial subtypes (Endo_C2_EndoBlood1, Endo_C3_EndoBlood2) with high expression of FLT1, CD34, PLVAP (Fig. 4a, b and Supplementary Fig. 11a)[38]. Here EndoBlood2 cells highly expressed tip-like markers such as RGCC, COL4A1and NOTCH4, whereas the expression levels of genes associated with stalk-like feature and immunity activation[19,36,39] such as ICAM1, HLA-DQB1and CCL2 were obviously increased in EndoBlood1 (Fig. 4c). In addition, when testing

**Fig. 2 | Characterization of functional gene modules from heterogeneous malignant tumor cells. a** Chromosomal landscape of inferred large-scale CNVs distinguishing malignant tumor cells from non-malignant epithelial cells in samples of T4-1 and T4-2. Amplifications (red) or deletions (blue) were inferred by averaging expression over 100-gene stretches on the indicated chromosomes. **b** t-SNE plot of 19207 malignant tumor cells colored by sample origins. **c** t-SNE plot of malignant tumor cells from RBT, NBT, and NAT colored by annotation groups. **d** Featured gene expression profiles of tumor cells across different samples from RBT, NBT, and NAT groups, separated by dashed vertical lines. **e** Heatmap depicting pairwise correlations of metagenes derived from 11 tumor samples from RBT, NBT and NAT groups. Clustering identified five underlying functional gene modules in malignant cells across samples. The dot size is proportional to the value of the correlation. Source data are provided as a Source Data file. **f–h** Boxplots of module score in each sample across RBT ($n = 3$), NBT ($n = 4$) and NAT ($n = 4$) groups for EMT_extended module (**f**), Cell-cycle module (**g**), and Epi_development module (**h**) separately, using scRNA-seq data. **i–k** Boxplots of module score in each sample across RBT ($n = 15$), NBT ($n = 15$), and NAT ($n = 7$) groups for EMT_extended module (**i**), Cell-cycle module (**j**) and Epi_development module (**k**) separately, using bulk RNA-seq data. *P* values were calculated by two-sided Student's *t* test, *P* values ≤ 0.05 are represented as *, and ≤0.01 as **; the centers of boxplots correspond to median values, with the boxes and whiskers corresponding to the corresponding interquartile ranges and 1.5× interquartile ranges; colored dots denote each samples; source data are provided as a Source Data file (**f–k**). **l–n** Survival analyses of 30 patients from treatment-naive in our HPC cohort with high or low expression scores of EMT_extended module (**l**), Cell-cycle module (**m**), and Epi_development module (**n**) separately. Patients were stratified by the mean module scores. *P* values were calculated by log-rank test.

the markers for TA-HECs critical for anti-tumor immunity by mediating lymphocyte entry into tumors[37], we observed EndoBlood1 had relatively higher expressions of its type-specific genes like LGALS3 and CTSC, along with other expression patterns similar to TA-HECs (Fig. 4c and Supplementary Fig. 11b). Further regulon[40,41] and signaling pathway enrichment analyses were done (Fig. 4d and Supplementary Fig. 11c), suggesting that EndoBlood1 was involved in pro-inflammatory and antigen presentation processes and Endoblood2 functioned in pro-tumor way by cell migration and angiogenesis. When pairwise comparing RBT with RAT and NBT with NAT, we found the proportions of cells with low StalkEC score but high TipEC score in Q4 decreased in both two conditions, from 55.84 (RBT) to 15.79% (RAT) for treatment-sensitive samples and from 51.52 (NBT) to 42.48% (NAT) for treatment-resistant samples (Fig. 4d). These results suggested the combined treatment could remodel the vascular endothelium in TME by reducing cells, which would be likely to migration and form new vessels.

## Two featured fibroblast subtypes identified in HPC

For a total of 9115 fibroblast cells, we also identified five cell subtypes, including one proliferative subtype (Fib_C1_proFib; MKI67, NUSAP1, PLK1), one myofibroblast (Fib_C2_MyoFib; ACTA2, PDGFA, CDH6) and three CAFs (Fib_C3_CAF1, Fib_C4_CAF2, and Fib_C5_CAF3; FAP, PDPN, PDGFRA) (Fig. 4e, f and Supplementary Fig. 12a)[42,43]. CAF1 highly expressed inflammatory CAF (iCAF) marker genes such as CFD, CXCL14, IGF1, while CAF2 and CAF3 showed similarly high gene expressions such as POSTN, CTHRC1, MMP14 and COL12A1 signatured by matrix CAF (mCAF) (Fig. 4g)[44]. Through the comparisons by stress-related genes such as MTIX and DDIT4[45,46] and pathway activation in extracellular matrix organization and collagen metabolic process, we further capture different signatures between CAF2 and CAF3 (Fig. 4h, i). Considering both the above differences and CAF3's unique sample origin (Supplementary Fig. 12b), we excluded CAF3 for downstream analyses. Consistent with previous research[47], the inferred developmental trajectory showed myofibroblast cells could evolve towards both tumor-promoting and tumor-suppressive directions (Supplementary Fig. 12c). When comparing the proportions of fibroblast subtypes among groups, we found CAF1 was abundant in RBT group and CAF2 was abundant in NBT group. Meanwhile, the percentage of CAF2 decreased from RBT to RAT, whereas the proportion was slightly increased from NBT to NAT, indicating CAF2 in TME could be better remodeled by effective combined treatment in treatment-sensitive groups of RBT and RAT (Fig. 4j and Supplementary Fig. 12d).

## Comparison of intercellular interactions among different groups in HPC

To characterize and compare intercellular interactions among RBT, NBT, and NAT groups in HPC, we inferred putative cell-to-cell interactions with CellPhoneDB from high-resolution scRNA-seq data[48,49]. Different cell cross-talk profiles were described among the three groups (Supplementary Fig. 13a). More intercellular interaction links existed between EndoBlood and CAF, as well as between CAF and CD4Treg in NBT. In contrast, the molecular interactions likely for tumor killing and antigen presentation between CD8T and malignant epithelial (MalignantEpi) cells, as well as DC, were more abundant in the RBT group (Fig. 5a and Supplementary Fig. 13b–e).

Interactional pairs related to specific biological functions were further assessed in detail. We observed more intensive interactions related to immunological mobilization (IFNγ-Type II IFNR, CD28-CD86, CD55-ADGRE5) in the RBT group than that in NBT (Fig. 5b)[11,50,51]. In contrast, MalignantEpi cells were predicted to interact with CD8T and CD4Treg through classical immune-suppressive pairs such as CD99-PILKα, PVR-TIGIT, and NECTIN3-TIGIT[11,52,53], which showed higher expression levels in NBT and NAT groups (Fig. 5c). As for function of angiogenesis represented by interaction pairs including LAMC1-A6b1, FN1-A3b1, ADRB2-VEGFB, KDR-VEGFC[54-58], the overall activation was lowest in RBT group and highest in NBT (Fig. 5d). Interactions related to lymphocyte recruitment signaling to exhibit anti-tumor effect between CD8 + T/NK and CAF(CXCR3-CXCL9[49], CXCR3-CCL19[59]) were most intensive in RBT group, while the pro-tumor state was most enhanced by interactions between Treg and CAF in NBT group (Fig. 5e). In addition, the cross-talk focusing on extracellular matrix (ECM) modeling (COL4A2-A2b1, COL1A2-a1b1)[60-63] was more abundant in NBT group than that in RBT group, and its strength was decreased in NAT group after treatment (Fig. 5f). Taken together, ligand-receptor interaction analyses suggested that RBT group had more favorable TME with more immunological stimulating signaling, while NBT group showed a complex state with high levels of angiogenesis, ECM remodeling and immunological inhibitory signaling.

## The classifier model trained to predict responses of the combined therapy in HPC

With both of single-cell and bulk RNA data in hand, we tried to take advantage of advanced tools like CIBERSORTx or BayesPrism to infer cellular compositions for further explorations[64-66]. Considering applicability of above tools to our detailed characterized cell subtypes, we finally used CIBERSORTx for extraction of subtype signature matrix from HPC scRNA-seq data, and then deconvolved our corresponding HPC cohort of bulk RNA-seq data to test whether there would exist a relationship between cellular compositions and treatment efficacy[65].

After the pre-test of input subtypes (Supplementary Fig. 14), we finally chose 15 well-characterized subtypes above to digitally estimate the non-malignant cell abundance via CIBERSORTx. We grouped 15 subtypes into two groups named "tumor-suppressive group" and "tumor-promoting group". The former group included cell subtypes of Endoblood1, CAF1, CD8T_naive, CD8T_cytotoxic, CD4Th, monocyte, pDC, moDC, and cDC1, while the latter consisted of EndoBlood2, CAF2, CD8T_exhaust, CD4Treg, cDC2 and macrophageM2 (Fig. 6a). Grouped by radioactive and histochemical diagnostic information, 44 samples from four groups showed different subtype composition profiles (Fig. 6a, b). Compared with samples in NBT, those in RBT had more

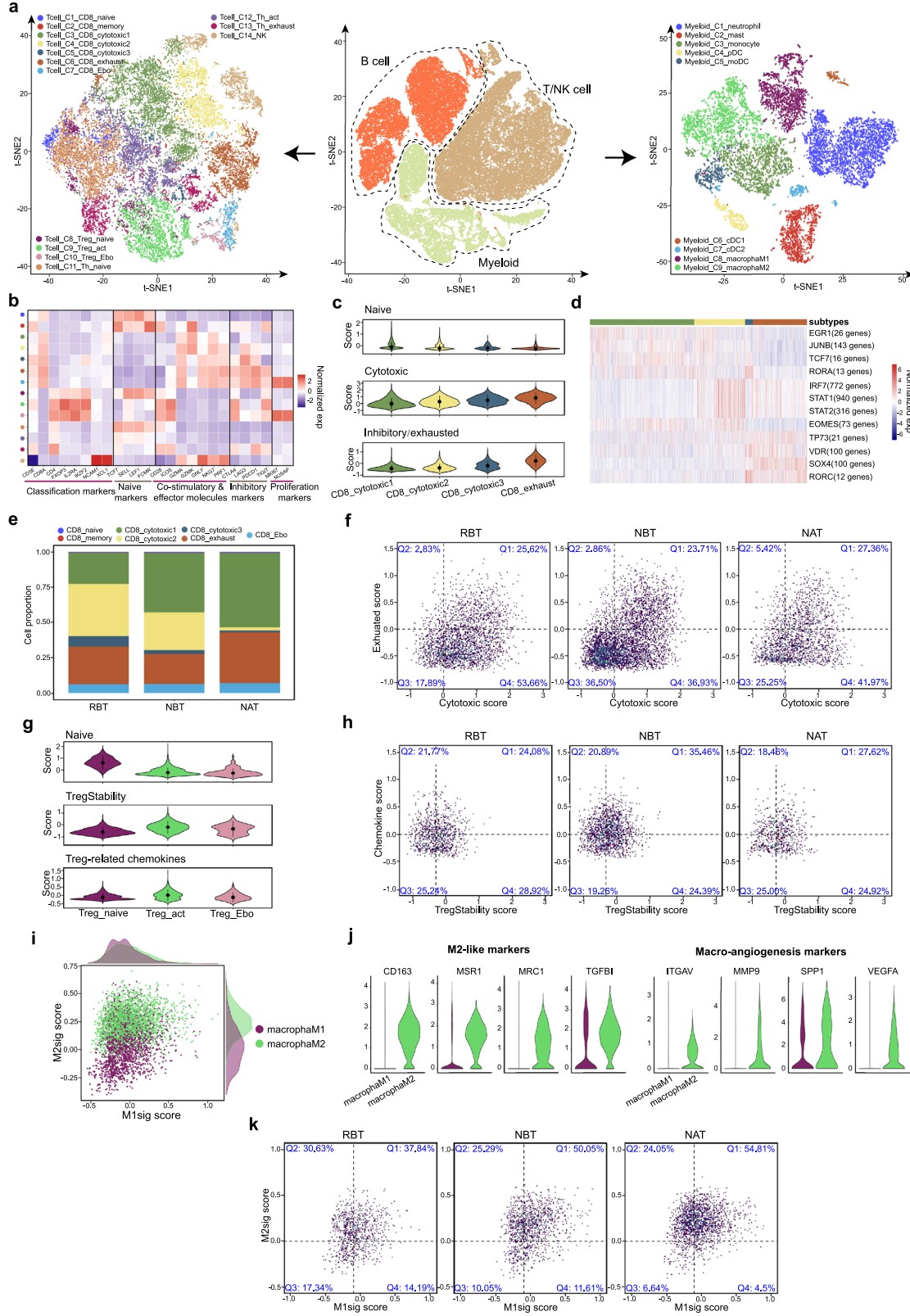

tumor-suppressive cells (58.7% vs 44.7%) and fewer tumor-promoting cells (41.3% vs 55.3%) (Fig. 6b). In the meanwhile, there was an increase of tumor-suppressive cells and decrease of tumor-promoting cells for samples in NAT group compared to those in NBT group, indicating the combined treatment could improve the state of anti-tumor activity in TME through cell type compositions. Subsequently, we explored the prognostic roles of these subtypes' signatures in HPC (Supplementary

Fig. 15 and Supplementary Table 4). Generally, high scores of tumor-suppressive and tumor-promoting subtypes' signatures had positive and negative correlations with survival, respectively.

Next, we tried to test whether subtype compositions in TME could be utilized to predict curative effects of the combined therapy quantitatively in HPC. Here, we used the matrix data of subtype compositions and treatment response labels from samples in RBT and NBT

**Fig. 3 | Detailed characterization of immune cells in TME of HPC. a** Identification of cell subtypes in immune cells. **b** Average gene expression heatmap of functional markers in T/NK subtypes. **c** Violin plots showing the expression scores of naive, cytotoxic, and inhibitory/exhausted signature gene sets of four CD8 + T-cell subtypes ($n = 9989$). Inside black points denote median values, and lines denote the corresponding interquartile ranges. **d** Heatmap showing the activity of TF regulons across four CD8 + T-cell subtypes. **e** Proportion differences of all CD8 + T subtypes among RBT, NBT, and NAT groups. **f** 2D density dot plots showing the changes of the cytotoxicity and exhaustion states of all CD8 + T cells among RBT, NBT and NAT groups. Cells were partitioned into four parts according to the mean scores of two states with quadrantal percentages shown in the plots. **g** Violin plots showing the expression scores of naive, TregStability and related chemokine signature gene sets

of three CD4 + Treg cell subtypes ($n = 4103$). Inside black points denote median values and lines denote the corresponding interquartile ranges. Source data are provided as a Source Data file. **h** 2D density dot plots showing the changes of the TregStability and chemokines states of all Treg cells among RBT, NBT and NAT groups. Cells were partitioned into four parts according to the mean scores of two states, with quadrantal percentages shown in the plots. **i** Scatter plot showing the scores of M1 and M2 signatures for each macrophage cell. **j** Violin plots showing the expression levels of different marker genes between subtypes of macrophageM1 and macrophageM2. **k** 2D density dot plots showing the changes of the M1 and M2 signature states of all macrophages among RBT, NBT, and NAT groups. Cells were partitioned into four parts according to the mean scores of two states, with quadrantal percentages shown in the plots.

groups, training a non-linear support vector machine (SVM) binary classifier model for prediction of the combined treatment response (Fig. 6c). The model had a relatively high prediction accuracy, with AUC = 0.86 tested in our HPC cohort (Fig. 6d). In order to validate the efficiency of the prediction model, we conducted a small-scale prospective trial with additional 12 treatment-naive samples, 7 samples of which were from RBT group (Supplementary Table 5). After the deconvolution for these samples, we checked prediction results by comparing SVM outputs with clinical true labels, and then found the overall correction rate was 10/12 (Fig. 6e). The favorable results provided an exciting expectation that we could use the model to assess the sensitivity of the combined treatment for advanced treatment-naive HPC patients, which would enlarge the possibility to preserve their laryngeal function before health condition deteriorated (Supplementary Fig. 16).

In addition, we explored the pipeline of deconvolution and prediction in NPC for further checking. Dissecting the public cohort of NPC with 88 samples, we could grouped them into 3 clusters by subtype composition profiles (Supplementary Fig. 17a), with decreasing percentages of tumor-promoting subtypes from group I to group III (Supplementary Fig. 17b). Moreover, group I had better progression-free survival rate than group III (Supplementary Fig. 17c), indicating the method captured certain underlying features for predicting tumor progression.

Furthermore, we proposed some potential therapeutic approaches for samples in NBT and NAT groups with an in silico exploration via Beyondcell[67]. Following its instructions, we subset all malignant tumor cells from RBT, NBT, and NAT, and used the drug sensitivity collection (SSc) for potential drugs finding. Due to the high heterogeneity of malignant cells, we could only get high-sensitivity drugs with median switch points (Supplementary Fig. 18). With drug sensitivity scores of malignant cells and drugs' mechanisms, it is believed that patients in NBT and NAT groups could benefit from RO-3306 and CAL-101, respectively (Fig. 6f, g). RO-3306, whose sensitive scores were relatively highest in NBT malignant cells, could block the cell cycle in the G2/M phase and induce apoptosis in cancer cells as a CDK1 inhibitor[68]. And CAL-101, also named as Idelalisib and sold under the brand of Zydelig, is a medication used to treat certain blood cancers and could be used as the secondary strategy for treatment-resistant HPC patients with further validations[69].

## Discussion

Combining high-resolution scRNA-seq data with bulk RNA-seq data, we not only described a single-cell landscape of clinical advanced samples for HPC, but also provided potential indicators for clinical prognosis and diagnosis (Fig. 7). On the one hand, we established the relationship between functional gene sets from malignant cells and tumor prognosis. Considering tumor heterogeneity, our data confirmed the notion that gene modules, rather than individual genes, are more robust and appropriate as underlying units for describing tumor transcriptional variability[10]. On the other hand, non-malignant cell compositions in TME deconvoluted from bulk RNA-seq data were

trained for a quantitative SVM model to predict the response of combined treatments with satisfactory correction rates.

In detail, we identified five characteristic functional gene modules from heterogenous tumor cells. The genes in Ribosome module were involved in almost all aspects of biological processes, making it difficult to restrict its application in a specific biological function. Similarly, the genes in the Immunity module were hard to apply in bulk RNA-seq data in the perspective of tumor cells, because they are also expressed by immune cells. Therefore, we mainly used gene sets from other 3 modules, namely "Epi_development", "Cell-cycle", and "EMT_extended", to establish the correlations with tumor prognosis. In addition, bulk RNA-seq data of our HPC cohort was dissected by the bioinformatic algorithm to obtain various cell compositions in TME and then trained for treatment response prediction. For this purpose, the way to separate biopsies for RNA-seq is important[70,71]. We applied the same criteria[72,73] to obtain samples with both tumor and non-malignant parts to ensure that the cell compositions in TME were representative. In the pre-test for CIBERSORTx deconvolution, we found that macrophageM1, mast cells, NK cells, and MyoFib contributed little in distinguishing sample differences (Supplementary Fig. 14). Therefore, we excluded them and used other cell subtypes to train and establish classifier model based on SVM machine learning algorithm, focusing the differences between RBT and NBT groups. The training result was satisfactory in limited samples with the correction rate at 10/12, suggesting its potential to provide therapeutical advice for HPC patients in the future. Although there existed differences in etiology and histopathology between NPC and HPC, considering the lack of various non-malignant subtypes in public single-cell datasets of NPC, we applied the signature matrix of conserved non-malignant cell subtypes from HPC to decouple and group public NPC data for validation of our methodology. Our analysis revealed the signatures of Endblood1 and CAF1 were also related to poor prognosis in NPC, and there existed less Endblood1 and CAF1 cells in group III, which had longer progress-free survival rates. The results suggested our method possessed the potential to predict tumor prognosis in NPC as well.

In this study, we combined the radiological information, scRNA-seq data, and a cohort with bulk RNA-seq data to establish a binary classifier model. Although the present classifier model showed favorable prediction results in small-size samples, it was built and tested in a cohort of predominantly male patients, and more HPC patients with gender differences were needed to enroll in prospective trials to improve and confirm the efficacy of the classifier model in treatment response prediction of the combined therapy. In addition, the combined treatment could cause double effects. On the one hand, activated gene expressions and enriched signaling pathways related to anti-tumor effects in NAT group, suggesting treatment lysed tumor cells to release tumor-specific antigens to activate anti-tumor cells. On the other hand, decreased immunological cell numbers were also observed, indicating the treatment killed immune cells simultaneously. Therefore, it's necessary for further exploration to test whether patients of NBT group would benefit from the treatment of

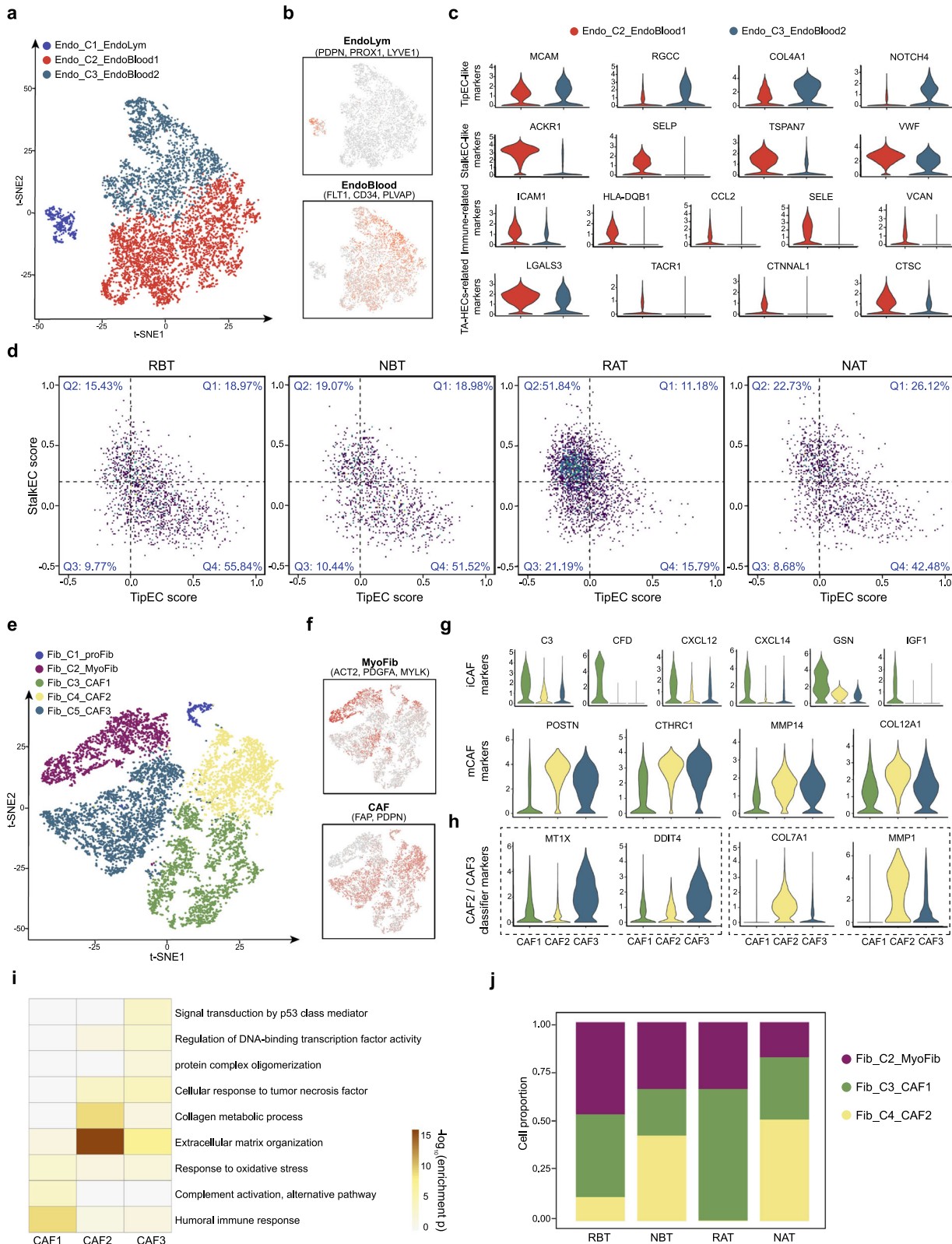

"chemotherapy plus immunotherapy", which would protect and enhance immunological functions by immunological drugs after tumor lysis caused by chemotherapy. In the future, the ideal application scenario is to divide advanced treatment-naïve HPC patients into two groups after their RNA-seq data are assessed by the classifier model. Patients tagged as "sensitivity" will be recommended to use the "classical combined therapy", and the others predicted as "resistance",

who have less potential to benefit from conventional therapeutical scheme, could choose clinical trials or other aggressive therapy to purse beneficial hopes (Supplementary Fig. 16).

In conclusion, our study identified certain functional gene sets to infer tumor prognosis, and established a quantitative classifier to predict responses of the combined therapy, both of which used only bulk RNA-seq and would be convenient and economic to provide

**Fig. 4 | Detailed characterization of endothelial cells and fibroblasts in TME of HPC. a** t-SNE plot of endothelial cells annotated into three subtypes. Each dot represents one single cell, colored according to cell subtype. **b** Normalized expressions of canonical marker genes to distinguish lymphatic and vascular endothelium. The depth of color from gray to red represents low to high expression. **c** Violin plots showing the expression levels of TipEC-like, StalkEC-like, immune-related, and TA-HECs-related markers in subtypes of Endoblood1 and Endoblood2. **d** 2D density dot plots showing the changes of the StalkEC and TipEC states of Endoblood cells among RBT, NBT, and NAT groups. Cells were partitioned into four parts according to the mean scores of two states, with quadrantal percentages shown in the plots. **e** t-SNE plot of fibroblast cells annotated into five subtypes. Each dot representing one single cell, colored according to cell subtype. **f** Normalized expressions of canonical marker genes to distinguish MyoFib and CAF. The depth of color from gray to red represents low to high expression. **g** Violin plots showing the expression levels of iCAF and mCAF markers in subtypes of CAF1, CAF2, and CAF3. **h** Violin plots showing the expression levels of classifier markers among CAF1, CAF2, and CAF3. **i** Heatmap showing the selected signaling pathways (rows) with significant enrichment of GO and KEGG terms for CAF1, CAF2, and CAF3. Source data are provided as a Source Data file. **j** Cell proportion differences of three focused fibroblast subtypes among RBT, NBT, RAT, and NAT groups.

diagnostic and therapeutical advice for advanced male predominate HPC patients.

## Methods

### Patient recruitment and sample collection

Eight male patients who were radiologically and pathologically diagnosed with advanced hypopharyngeal carcinoma (HPC) were enrolled in this study between December 2019 and January 2021. Totally 15 fresh clinical samples were obtained from the patients (Supplementary Tables 2 and 3), followed immediately by single-cell preparation as described below. Additionally, 44 HPC samples of our HPC cohort and another 12 HPC samples for respective research were collected for bulk RNA-seq profiles. Above 56 samples sent for bulk RNA-seq were derived from 56 individual patients (with 3 female patients) between July 2016 and August 2022. According to the previous statistics on the difference in the late-life incidence of HPC between men and women[4], the gender rate of our samples was roughly consistent, so we did not take sex and gender into account in our study. In addition, this information was reflected in the title, abstract and throughout the manuscript to avoid ambiguity. All patients' clinical characteristics are summarized in Supplementary Tables 1, 2, and 5. All the above clinical samples were collected at the Department of Otolaryngology Head and Neck Surgery, Beijing Tongren Hospital. Written informed consent was obtained from all participants for sample collection and analysis as well as for publishing-related information such as gender, age and TMN stage in necessary scientific researches. Ethical approval was obtained from the Ethics Committee of Beijing Tongren Hospital, Capital Medical University (TRECKY2016-025 and TRECKY2021-049).

### Collection and preparation of samples for bulk RNA-seq

The specimens with only tumor cells would lead to the loss of non-malignant cells, and the core part of tumor, occupied with necrotic cells, would cause low RNA-seq quality. It is a more recommended method to take the junction covering both tumor and adjacent samples, which could help to obtain tumor cells and other cell constitutions in TME together. For bulk RNA-seq, specimens obtained from Tongren hospital were subjected to total RNA isolation using a commercial RNA extraction kit (Takara). After whole-transcriptome amplification, library construction was performed using the Truseq RNA Library Prep kit v2 (Illumina) following the manufacturer's recommendations. Samples were sequenced using the Illumina HiSeq 2000 platform to generate 150-bp paired-end reads.

### Preparation of single-cell suspensions for droplet-based 10x scRNA-seq

The samples for scRNA-seq were also collected as above description and washed with phosphate-buffered saline (PBS; Solarbio), placed on ice, cut into small pieces (<1 mm³) and transferred to 5 mL Dulbecco's modified Eagle's medium (DMEM; Thermo Fisher Scientific) containing collagenase IV (1 mg/mL) (Thermo Fisher Scientific), DNase I (20 U/mL) (Invitrogen), Hyaluronidase (0.1 mg/mL) (Merch), and Dispase (1 mg/mL) (Gibco). The samples were transferred into gentleMACS C tube (Miltenyi Biotec), and ran h_TDK_3 program according to User Manual of MACSmix Tube Rotator (Miltenyi Biotec) and then filtered twice using a 40-μm nylon mesh (Thermo Fisher Scientific). After centrifugation (500×*g*, 4 °C, 5 min), the pelleted cells were suspended with ice-cold red blood cell lysis buffer (Solarbio) and filtered with a 40-μm nylon mesh. Last, the pelleted cells were suspended with 1 ml of Dulbecco's PBS (Solarbio), and the concentrations of live cells and clumped cells were determined using an automated cell counter (Luna fl). During the dissociation procedure, the cells were kept on ice whenever possible, and the entire procedure was completed in <90 min (generally ~70 min) to avoid the dissociation-associated artifacts recently described. A positive signal for a dissociation signature that reflects dissociation-associated changes in gene expression was obtained in <1% of the cells. Cell count and cell viability were measured before library construction and deep-sequencing, which was performed on Illumina NovaSeq 6000 by Annoroad Gene Technology Co., Ltd.

### Multiplex IHC staining assays

Multiplex IHC staining assays were performed on 4-mm-thick, formalin-fixed, paraffin-embedded slides using an Opal multiplex IHC system (NEL811001KT, PerkinElmer) according to the manufacturer's instructions. Briefly, after slide preparation and heat-induced epitope retrieval, slides were blocked with PerkinElmer Antibody Diluent Block buffer. In all, 100 μl antibodies were used after dilution as follows: anti-CD31 (CST no. 3528, 1/300), anti-FAP (Abcam, no. 207178, 1/100), anti-EPCAM (Abcam, no. 223582, 1/100) and anti-CD45 (CST, no. 13917, 1/400). Each slide was baked in the oven at 75 °C for 1 h. Then, the slides were deparaffinized with xylene and rehydrated through a graded series of ethanol solutions. After antigen retrieval in a microwave, the slides were washed in TBST wash buffer. After blocking, the sections were incubated with primary antibodies for 1 h and then incubated with 100 μl polymer HRP Ms+Rb as the secondary antibody (GT no. GK600711-B) for 10 min at room temperature. Opal fluorophores were pipetted onto each slide for 10 min at room temperature, and the slides were microwaved to strip the primary and secondary antibodies (Step 1). Then, we repeated the same protocol using the next primary antibody targets (Steps 2–6). Finally, DAPI was pipetted onto each slide for 10 min at room temperature (Step 7). The slides were covered with VECTASHIELD, and images were taken using a Vectra Polaris automated quantitative pathology system. The images were analyzed by inForm 2.3.0 software (PerkinElmer, Waltham, USA).

### Process of a small-scale prospective trial of additional 12 HPC samples

Overall, 12 specimens from treatment-naive HPC patients were performed for bulk RNA-seq to obtain subtype compositions in TME by CIBERSORTx after first CT scanning. Then they were predicted as sensitive or resistant sample by the classifier model (denoted as predicted labels). Then, the patients performed the second radiological test after one cycle of the combined therapy, and the changes of tumor mass were confirmed by comparing two radiological results to get the true labels of these samples.

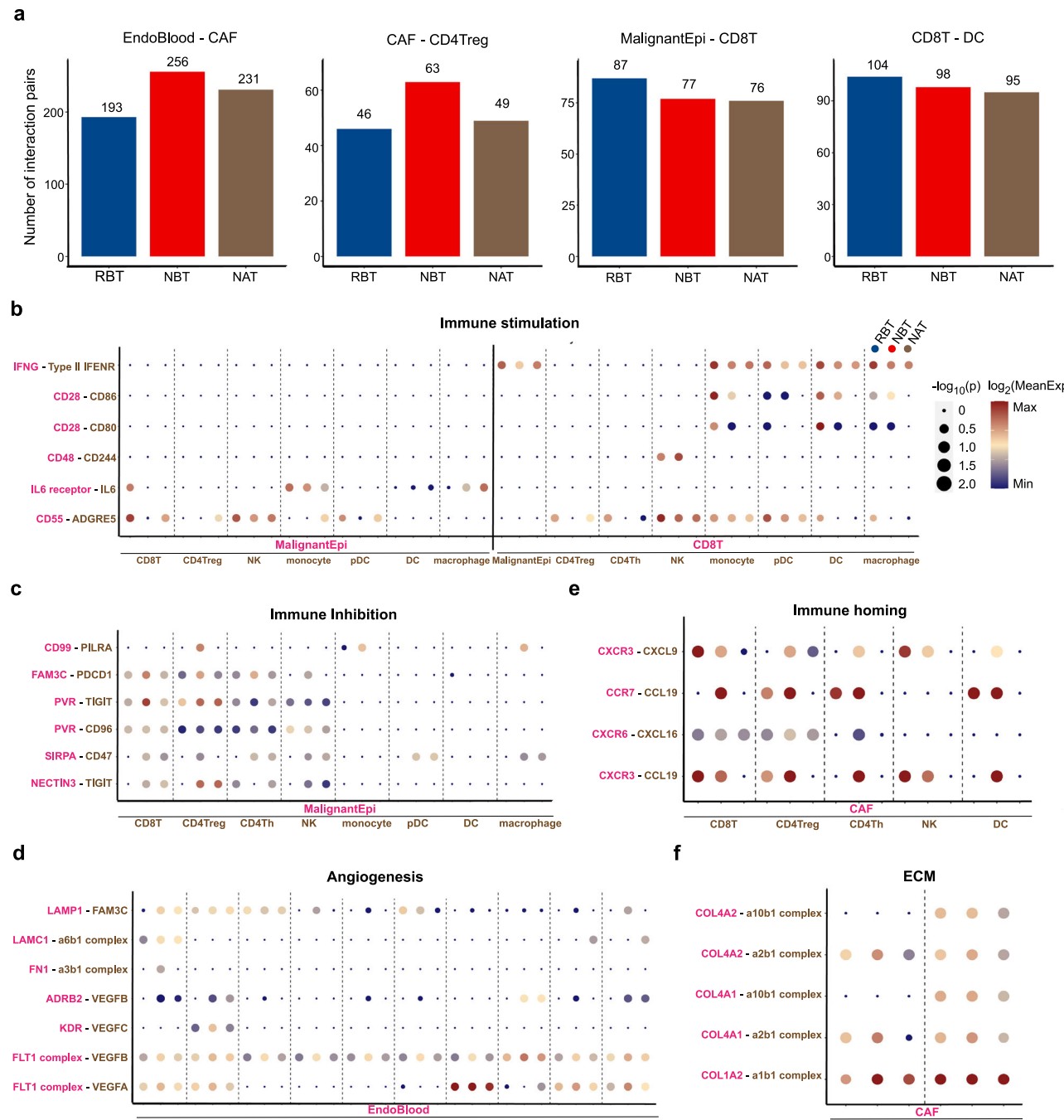

**Fig. 5 | Comparison of intercellular interactions among three groups in HPC.** **a** Bar plots showing different intercellular interaction numbers among three groups for four cell type pairs, including EndoBlood and CAF, CAF and Treg, malignant epithelial (MalignantEpi) cells and CD8 + T cells, as well as CD8 + T cells and DCs. **b**–**f** Selected ligand-receptor interaction profiles for five important biological functions, including immune stimulation (**b**), immune inhibition (**c**), angiogenesis (**d**), immune homing attracted by CAF (**e**), and extracellular matrix (ECM) modeling (**f**).The dot color from dark blue to dark red indicates the level of interaction. *P* values are presented by circle size (getting from one-sided permutation test). The means of the average expression levels of interacting molecule 1 in cluster 1 and interacting molecule 2 in cluster 2 are indicated by color. Source data are provided as a Source Data file.

## Single-cell gene expression quantification and removal of unqualified events

We used CellRanger (version 4.0.0) to generate a raw gene expression matrix for each scRNA-seq sample. Then as shown in Supplementary Fig 2a, quality control (QC) filters of scRNA-seq data consisted of basic and detailed parts. In the Basic QC, filtering of cells was firstly performed to remove the ones with <201 or >7500 expressed genes and with more than 25% unique molecular identifiers (UMIs) derived from the mitochondrial genome by Seurat R

package (version 3.2.2)[74,75]. Multiple primary-filtered expression matrices were directly merged with merge() function embedded in Seurat package. After the typical data process for scRNA-seq data, six major cell types were identified with featured markers. Next, we extracted cells from each group one-by-one for detailed QC. In this part, we comprehensively considered the characteristics of doublet event, as well as bad effects from dissociation, cell cycle, and contamination. We used package Scrublet and DoubletDecon to infer cell doublets[76,77] and checked the profiles of dissociation and cell-

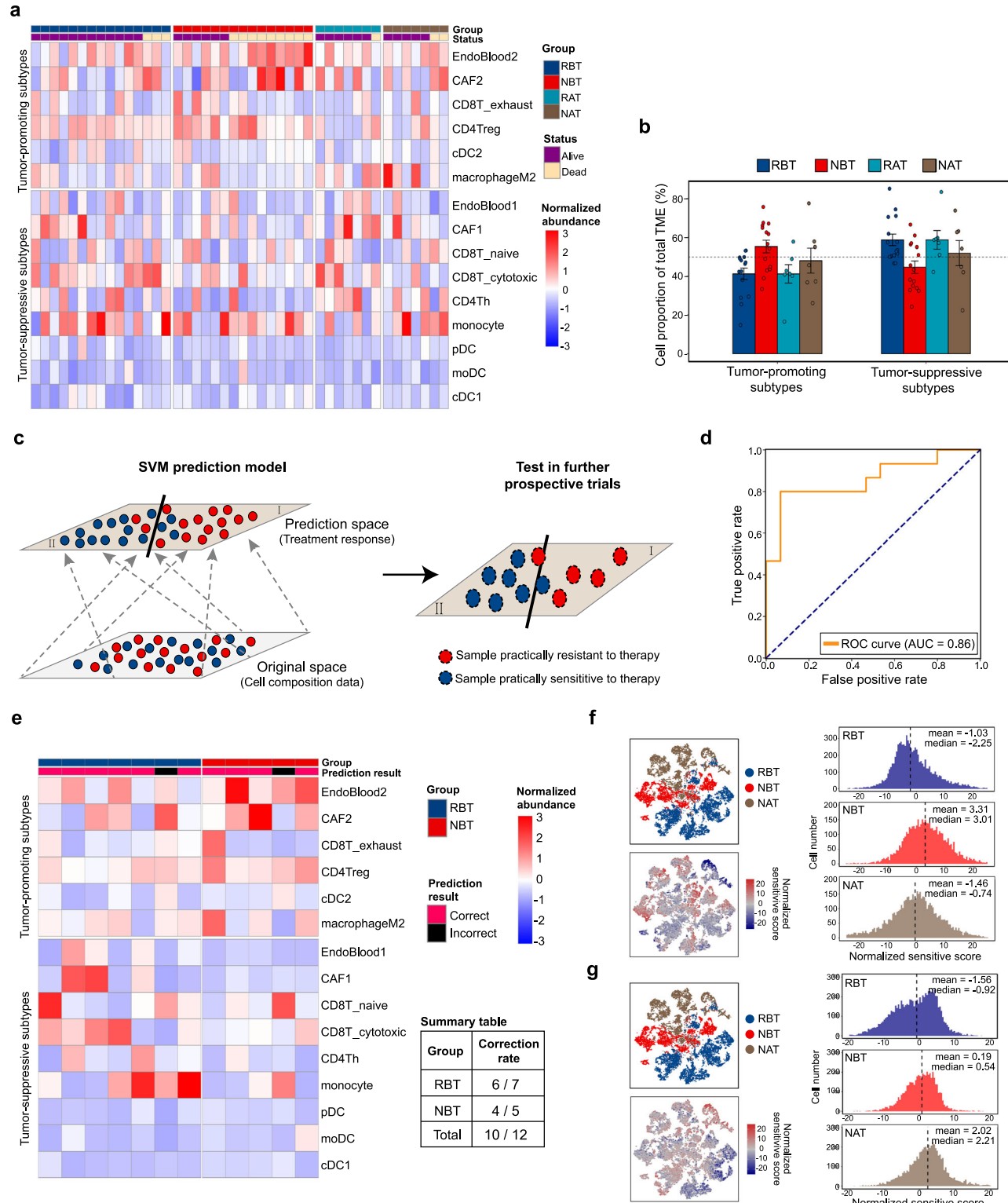

cycle with histograms and reduction t-SNE plots according to related researches[75,78]. In our study, we specifically found some contaminated cells, which had dual features from definitely two different cell types and mainly originated from the samples with relatively low cell viability in the measurement before sequencing. Finally, all qualified cells were obtained and merged for downstream analyses.

### Identification of the major cell types and their subtypes

For all qualified cells, gene expression matrices were log normalized to total cellular read counts and mitochondrial read counts by linear

regression implemented using the ScaleData() function embedded in Seurat package. Major cell types were annotated to known cell lineages using well-recognized marker genes with projection in the two-dimensional t-SNE representation.

For the identification of subpopulations for each major cell type, we repeated the above-mentioned steps, including normalization, dimensionality reduction, and clustering. We adjusted and checked the resolutions of clustering repeatedly according to averaged expressions of feature genes from literatures with help of single-cell auto-classification software SingleR (version 1.0.6)[79].

**Fig. 6 | Prediction of the combined treatment response based on non-malignant subtype compositions. a** Heatmap of the normalized cell abundance with fifteen subtypes estimated via CIBERSORTx and clinical records in our HPC cohort. All 44 samples from RBT, NBT, RAT, and NAT groups were deconvolved for estimation. **b** Average cell compositions in TME among four groups by dividing fifteen subtypes into two groups named as tumor-promoting subtypes and tumor-suppressive types. Error bars represent standard errors of cell constitutions in corresponding groups. The biological independent sample numbers in the four groups were 15, 15, 7, and 7, respectively. **c** Cartoon plot illustrating the processes for SVM construction from our HPC cohort and application for prospective trails in HPC. **d** Measurement for the prediction performance of SVM classifier model. The area under receive operating characteristic curve is 0.86 on the training samples. **e** Heatmap of the normalized cell abundance with 15 subtypes for additional 12 samples. Tag of correct represented the labels from prediction model and true

clinical result were identical, while tag of incorrect represented the labels from the prediction model and true clinical result were different. A summary table was shown to summarize the separate and total correction rates. Source data are provided as a Source Data file. **f** Drug sensitivity of RO-3306 in malignant tumor cells among RBT, NBT, and NAT groups. t-SNE plots of tumor cells annotated into three groups (left top) and colored by normalized sensitivity scores (left bottom). Histogram plots show the distribution of normalized sensitivity scores for tumor cells in three groups, with dashed vertical lines representing the corresponding median scores. **g** Drug sensitivity of CAL-101 in malignant tumor cells among RBT, NBT and NAT groups. t-SNE plots of tumor cells annotated into three groups (left top) and colored by normalized sensitivity scores (left bottom). Histogram plots show the distribution of normalized sensitivity scores for tumor cells in three groups, with dashed vertical lines representing the corresponding median scores.

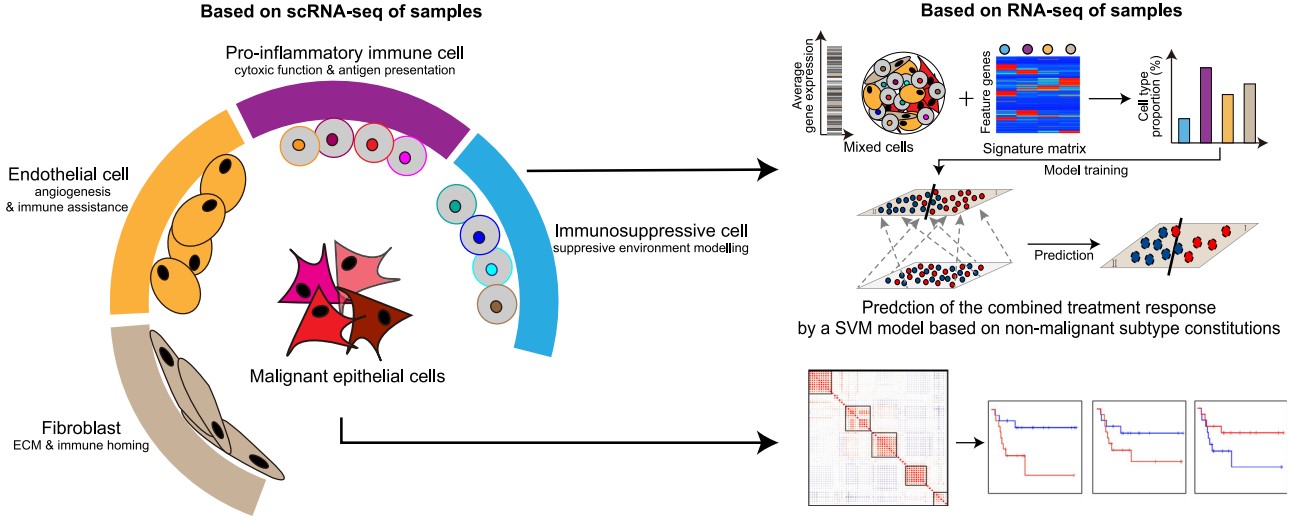

**Fig. 7 | Schematic diagram of single-cell landscapes in HPC for analyses of prognosis and treatment response.** The complex landscape of TME in HPC was illustrated by high-solution scRNA-seq data, including heterogenous malignant tumor cells and various non-malignant cell types, which functioned in their specific ways. On the one hand, we established the relationship between functional gene

sets from malignant cells for tumor prognosis inference. On the other hand, non-malignant cell compositions in TME deconvoluted from bulk RNA-seq data were trained for a quantitative SVM model to predict the response of combined treatments for advanced HPC patients with satisfactory correction rates.

## CNV analysis and identification of malignant epithelial cells
To identify malignant epithelial cells, we identified evidence for somatic alterations of large-scale chromosomal copy-number variants (CNV), either gains or losses, in a single cell using inferCNV software (https://github.com/broadinstitute/inferCNV). The raw single-cell gene expression data of epithelial cells in each sample was extracted from the Seurat object for testing. The single-cell data of epithelial cells from NT group were used as reference cells. We preformed inferCNV analysis with the default parameters.

## Identification of functional gene modules embedded in heterogenous malignant tumor cells and extraction of the corresponding gene sets
Focusing on the malignant tumor cells from RBT, NBT, and NAT groups, NMF was used to identify expressed functional gene modules. Using the NMF R package (version 0.23)[23], we applied NMF to the normalized gene expression matrix of each sample, in which genes with standard deviations of expression <0.5 were excluded. We selected five or six as the factorization parameter (rank) according to cophenetic correlation coefficients in corresponding samples after pre-test for rank selection (Supplementary Fig 5a) and used extractFeatures() function for genes extraction of each meta-signature. Finally, a total of 61 metagenes were identified across the 11 tumors.

The all metagenes were used for calculation of module scores in malignant cells, and then compared by Pearson correlation before further clustering. Five clusters of biological modules were identified manually. For each module, we extract genes from meta-signatures that commonly expressed in at least three samples with considering their function in early researches. Finally, we got about 20 feature genes for each module (Supplementary Table 4).

## Pseudotime trajectory analysis
We applied the Monocle2 R package to determine the potential development lineages in the T cell, B cell, myeloid, and fibroblast subpopulations[80]. The differentially expressed genes across the clusters were identified by dispersionTable() function in Monocle2 with default filtering parameters. The cells were ordered in pseudotime, where the best trajectory tree was fit after the reduction of dimensionality of the data by Reversed Graph Embedding algorithm.

## SCENIC analysis
SCENIC analysis was conducted with the pySCENIC package (version 0.9.9)[26], a lightning-fast python implementation of the SCENIC pipeline. Two gene-motif rankings (10 kb around the transcription start site (TSS) or 500 bp upstream of the TSS) were used to determine the

search space around the TSS, and the 20-thousand motif database was used for RcisTarget and GENIE3.

## Characterization of functional scores in single-cell data

To evaluate the potential biological functions of interested cells, we calculated the scores of functional modules using the AddModuleScore() function in Seurat at the single-cell level and averaged at sample level if needed. The functional modules including five signature programs for malignant cells, naive, cytotoxic and exhausted scores for CD8 + T cells, naive, TregStability and Treg-related chemokine scores for CD4 + FOXP3 + Treg cells, M1, and M2 signature scores for macrophages, as well was TipEC and StalkEC scores for vascular endothelial cells. The involved gene sets are listed in Supplementary Table 4. The calculating scores were used for comparisons of cell subtypes or changes among different treatment groups.

## Pathway enrichment analysis

To gain functional and mechanistic insights between cell subtypes, we performed Gene Ontology (GO) and KEGG Pathway enrichment analyses using Metascape (http://metascape.org/) to identify biological pathways that were enriched in a certain gene list more than that would be expected by chance[81]. The gene lists were calculated with lnFC >0.20 in clusters and greater than 15% expression threshold. To compare the difference of signaling pathway enrichment in malignant cells between samples, we performed the gene set variation analysis (GSVA, version 1.34.0) using the selected molecular signatures[82], including hallmark pathways, Gene Ontology (GO) and KEGG Pathways from MSigDB database.

## Cell–cell communication analysis via CellPhoneDB

CellPhoneDB (version 2.0.6) used the cluster annotation and raw counts from our single-cell transcriptomics data to compute cell–cell communication within the cell subtypes[49]. The default ligand-receptor pair information was used in this process with considering only ligands and receptors with expression in more than 15% of the cell subtypes. The P values were calculated at 1000 times permutation test, and values greater than 0.05 indicated significant enrichment of the interacting ligand-receptor pair in each of the interacting pairs of cell subtypes.

## Analysis of bulk RNA-seq data of our HPC cohort and public NPC cohort

Pair-end reads with high quality of 56 samples (44 samples with 12 additional ones for prospective research) were aligned to the human genome (GRCh38) using HISAT2 (version 2.1.0) with default setting. Software featureCounts (version 2.0.3) was used to quantitate the read counts of each gene in samples. The expression levels of genes were normalized by gene length and sequencing depth with edgeR (version 3.28.1) among samples. In group comparisons of malignant modules, GSVA R package (version 1.34.0) was used to calculate module scores of each sample with defined gene sets[82].

The data of public NPC cohort was retrieved from the public GEO database (GSE102349), including 113 NPC tissue samples profiled by bulk RNA-seq. However, only 88 samples with clinical progression information were used in this study. We downloaded the expression matrix from databased and filtered the genes that did not express in at least 50 samples for further analysis.

## Survival analysis with gene expression signatures

The expressions of functional modules or signatures for specific cell subtypes were evaluated by GSVA R package (version 1.34.0). To assess the prognostic values of gene/module expressions, samples from our HPC cohort were allocated into two groups with high and low levels of specific features by in mean or median way. Kaplan–Meier survival curves were plotted with the Survival R

package to show differences in survival time and evaluated by the two-sided log-rank test.

## Estimation of cell abundances from bulk RNA-seq data via CIBERSORTx

Based on our single-cell sequencing data, we selected interested and representative subtypes for generating the signature matrix. With the reference, CIBERSORTx (version 1.1.0) (https://cibersortx.stanford.edu/) deconvoluted bulk RNA-seq data, including both of our HPC and public NPC cohorts into the subtype compositions in each sample using the S-mode batch correction.

## Construction of a SVM model for predicting responses of the combined therapy in HCP

In order to harness cell compositions from our HPC cohort for the prediction of responses to combined treatments, we trained a support vector machine (SVM) based on 30 treatment-naive samples, half of which were pre-defined as RBT group. To get the SVM classifier, we first performed principal component analysis (PCA) on the training samples and seven top principal components were selected for data transformation. The SVM classifier was derived from Python scikit-learn module and the non-linear sigmoid kernel was chosen with regularization parameter set as 1.5. The training error and fivefold cross-validation error of the SVM classifier were 0.17 and 0.25, respectively, and the area under receive operating characteristic curve (AUROC) is 0.86 on the training samples. Then we evaluated the SVM classifier performance on another 12 HPC samples as the test dataset, which was composed of 7 RBT samples and 5 NBT samples.

## Statistics and reproducibility

HPC is a rare malignancy. Totally we used 15 samples for single-cell RNA-seq analyses and 56 samples for bulk RNA-seq analyses. No statistical tests were performed for sample size calculation but it was sufficient for this proof-of-concept study corroborated by two kinds of data. All criteria for data exclusion were established and described as above for quality control. All HPC patients were recruited randomly in this study and divided into groups according to their clinical diagnosis. Investigators were blinded to patient identity only with coded sample ID. All statistical analyses and presentations were performed using R (http://www.r-project.org). All Data points were shown for bar plots and boxplots with a sample size ≤10. For larger sample sizes, box and violin plots were used to visualize the data distribution. Data were presented as the mean values ± SE in bar plots. P values were evaluated by two-sided Student's t test, one-sided permutation test and log-rank test. P values > 0.05 were considered not statistically significant and represented as ns., P values ≤ 0.05 were represented as *, ≤0.01 as **. Multiplex IHC staining assays were confirmed in three biological replicates.

## Reporting summary

Further information on research design is available in the Nature Portfolio Reporting Summary linked to this article.

# Data availability

The raw sequence data generated from bulk and single-cell RNA-seq of clinical samples in this study have been deposited in the Genome Sequence Archive (Genomics, Proteomics & Bioinformatics 2021) in National Genomics Data Center (Nucleic Acids Res, 2022), China National Center for Bioinformation/Beijing Institute of Genomics, Chinese Academy of Sciences, under accession number HRA003383. The data are available under restricted access for relevant data protection regulations considering the data contains human genetic information, and the access can be obtained after being authorized by its Data Access Committee (DAC) by checking the identity and purpose of applicants. Generally, reasonable requests will be approved within

2 weeks, and the download permission will be opened. The publicly available NPC bulk RNA sequencing data used in the study are available in GEO with the accession number GSE102349[22]. The patient information for single-cell sequencing and bulk RNA sequencing is available in Supplementary Tables 1, 2, 3, and 5. Source data are provided with this paper.

## Code availability

The scripts are available at https://github.com/Sara0201Tao/2022HPC[83].

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

## Acknowledgements

This study is supported by the National Key Research and Development Program of China (2019YFA0906103) [Z.X.], National Natural Science Foundation of China (No. 61721003 [Z.X.] and No. 82072997 [Y.Z.]), and Beijing Municipal Science & Technology Commission (No.Z221100007422045) [Y.Z.]. We thank the department of Pathology of Beijing Tongren Hospital for assistance with mIHC staining and schematic drawing, and we thank the department of Radiology of Beijing Tongren Hospital for radiological images collection and radiological diagnosis. We are grateful to the members of the MOE Key Laboratory of Bioinformatics and Bioinformatics Division of Tsinghua university for helping to establish the pipeline for biological informative analyses. We appreciate the technical support provided from Beijing Syngentech Co., Ltd, Annoroad Gene Technology Co., Ltd, and Beijing Biological Data of Human Co., Ltd.

## Author contributions

Z.H., Z.X., and G.L. conceived this project. Under the supervision of Z.H. and Z.X., Y.Z., G.L., W.G., G.Y., W.G., and L.F. performed experiments. Under the supervision of G.L. and Z.X., M.T. performed bioinformatics analyses, and H.N. gave useful suggestions in the process of SVM model construction. J.G. provided valuable advice on the quality control of scRNA-seq data and the layout of the manuscript. G.L., M.T., Y.Z., Z.X., and Z.H. wrote the manuscript.

## Competing interests

Two patents based on the study were submitted by Z.X., Y.Z., G.L., and M.T., which were entitled as "Featured gene sets based on scRNA-seq profiles of malignant tumor cells for the prognosis prediction of hypopharyngeal carcinoma" (application number, no 202310103758.7) and "A prediction method of the therapeutic efficacy for the combined treatment in advanced hypopharyngeal carcinoma based on the integration of single-cell and bulk transcriptome profiles" (application number, no 202310106947.X). The remaining authors declare no competing interests.
