## [Peer review file · Nature Communications]

REVIEWER COMMENTS

Reviewer #1 (Remarks to the Author): Expert in head and neck cancers

Y. Zhang and coauthors collected HPC tumor samples before and after combination therapy and classified tumor samples into four groups (NaiveGood, NaiveBad, TreatGood and TreatBad) based on radiological diagnosis. The authors performed single cell RNA-seq to analyze HPC tumor microenvironment, and identified gene modules and cell constitutions that affect the response to combination therapy. Based on the single-cell HPC TME map, the authors deduced cell constitution from the bulk RNA-seq data of the tumors, and established a classifier model to predict the efficacy of combination therapy.

Overall, this study provides a useful resource of single-cell RNA before and after the standard clinical treatment of the rare but malignant HPC tumor, which may promote the understanding of HPC tumor progression and have the potential to improve the treatment practice. Moreover, this study shows an interesting and useful strategy for identifying prognosis indicators for tumor treatment based on single-cell and bulk RNA-seq data. I have several comments to which the authors should address:

1. To predict the efficacy of the combination therapy, the authors trained a SVM model using a set of HPC bulk RNA-seq data with a ROC curve of 0.86, suggesting that this model has the potential to serve as a prognostic tool. Although the HPC is a rare tumor, I encourage the authors to collect another set of HPC biopsy samples to further validate the efficiency of the SVM model.
2. Different cell types were identified in the HPC tumor samples using single-cell RNA-seq. It will be more convincing to validate these results by using immunohistochemical staining method, at least for T lymphocytes, macrophage, fibroblasts and endothelial cells.
3. Recently, A Asrir et. al (DOI: 10.1016/j.ccell.2022.01.002.) reported that tumor-associated high endothelial venules (TA-HEVs) co-express MECA-79+HEV sialomucins and E/P-selectins, which are involved in the tumor prognosis. Are TA-HEVs similar to the endothelial cells (Endoblood1 and Endoblood2) found in this study?
4. The data processing and quality control are important for single-cell RNA-seq analysis. In the method, the authors described the 10x raw data process based on the number of detected genes, UMIs and doublets. How about other factor, such as cell-cycle related genes?; and what are the contaminated cells mentioned in Page 14 line 469?
5. The authors should clarify how the authors used non-negative matrix factorization (NMF) to identify 61 megagenes from 11 HPC samples shown in Figure 2e.
6. In the section of TME compositions and cell-cell interactions in HPC, cellular interactions between 12 conventionally defined cell types were shown in Figure 5. However, more subtypes were defined in

Figure 3. Does the study of the cell-cell interactions between these subtypes provide more information on TME?

7. I strongly suggest to have professional editor to proofread the English throughout the manuscript before it is accepted for publication.

Reviewer #2 (Remarks to the Author): Expert in single-cell RNA-seq, tumour microenvironment, and drug response prediction

Zhang et al. uses single-cell transcriptomics to comprehensively dissect the tumour microenvironment in hypopharyngeal carcinoma (HPC) biopsies before and after combined therapy and identified three gene modules associated with tumor progression.

Bulk RNA-seq data from HPC patients is also used for the analysis of tumor gene modules to predict prognosis. Finally the authors propose a classifier to predict the relationship between cell subtype abundance in TME and the efficacy of combined treatment.

In general the manuscript is well written, fluent, clear and understandable. The authors make a great effort performing a very comprehensive dissection of TME cell subpopulations and its functional implications according to the HPC subgroups based on clinical diagnosis (NG,NB,TG,TB).

In addition, currently there are not too many single-cell datasets including cancer patients before and after combined therapy. As far as I know, there is no other published work that has presented single-cell results in HPC and with such a complete description of the cell types present in HPC microenvironment. Undoubtedly, such a single-cell dataset is of interest to the cancer research community.

Major points:

- In the Discussion the authors state “In the future, the ideal application scenario is to divide treatment-naïve HPC patients into two groups after their RNA-seq data are analyzed by the classifier model. Patients tagged as “NaiveGood” will be recommended to use the “classical combined therapy” plan, and

the other group predicted as “NaiveBad”, who have less potential to benefit from conventional therapeutical scheme, could choose clinical trials or other aggressive therapy to pursue beneficial hopes.”

I believe that the authors could go a step further and propose some treatments through an in silico prescription study. In other words, this study should be enriched by an analysis of in silico therapeutic approaches (see PubMedID: 34911571) for the 4 groups of HPC, especially for those subtypes that do not respond well to combination therapy (i.e. NaiveBad, Treated Bad if possible). Such a study would allow to suggest potential candidate treatments to be proposed for patients who do not respond to combined treatment. In addition, it would allow to know whether or not the greater functional heterogeneity observed in the TB group (line 142, Suppl Fig 5) corresponds to greater therapeutic heterogeneity.

In my opinion, it could be very interesting and would add more value to the work.

- The authors must also upload both scRNA-seq and RNA-seq HPC datasets to a publicly accessible repository to make them available to the scientific community.

Minor:

-Perhaps if a small table or description of the number of patients included in the paper, the type (NG, NB, TG, TB) and the type of sample (i.e. single-cell, bulk) could be included in one of the main figures. It would clarify the reading of the paper and summarise what is described by the authors from line 104 onwards.

- Please include legend in Figure 1h

- Figure 5a. It is almost the same Figure for NG, NB and TB. The authors should consider moving it to Supplementary. In addition, there is a typo in the header of the Circos plot, “NaiveBad” is repeated.

- Figure 5b has no legend

- Figure 6a: The columns are not sorted consistently according to the Status label (Alive and Dead). Please, sort them consistently.

- Typo in Figure 7: "Geneset expression".

- Given that the authors have both single-cell and bulk data, they may find it interesting to use in future studies other tools that take advantage of both types of data to infer cellular composition and expression in large cohorts of bulk RNA-seq data. (e.g. PMID: 35469013)

Reviewer #3 (Remarks to the Author): Expert in head and neck cancer pathology and therapy, and tumour microenvironment

The overall goal of this study was to identify the relationship between TME cell type composition and patients' response to combined therapy. Overall, this is a very interesting approach to investigating hypopharyngeal cancer with potential clinical implications. However, it is important that the following are addressed: Is assessment of the cell constitutions in TME better than the current approach for therapy selection? Additionally, it is unclear why the authors extrapolated data from hypopharyngeal cancer to nasopharyngeal cancer which differs in location, histopathology, and anatomic site. There are also several minor discrepancies between data labels in the Results section and figures.

Major concerns.

1. In the Kaplan-Meier curves, (2h, 2k, 2n), the number of patients at risk at each time point should be shown to facilitate data interpretation. How does survival compare between groups, i.e. EndoBlood2, CAF2 etc (log rank tests)? How do independent variables affect the survival time.
2. In using the cell constitutions in TME (Figure 6) to predict curative effects of combined therapy, is this better than the current approach? What is the magnitude of the difference between groups?
3. Supplementary Figure 8e: Contrary to what is reported in the results, cDC2 appears to be more of a killer subset than pDC.
4. Supplementary Figure 9: The resolution of this figure is poor; the small print that presents the key of each figure is very difficult to read.
5. Figure 4c: The authors report high expression of NOTCH4 in C3_EndoBlood2. This should be shown in the figure and labeled accordingly. Others mentioned in the results such as TSPAN7 are also not shown; ICAM1 is incorrectly identified as ICMA1 in the results.
6. Fig. 4f: The authors state that "Compared to the NB group, the TB group showed decreased cells with lower StalkEC scores but higher TipEC scores (Fig. 4f; Supplementary Fig. 10b, c), indicating the combined treatment could trigger antitumor effects by remodeling the vascular endothelium to decrease cell migration." The decrease in StalkEC is only ~2%; is this significant?
7. Please check labels throughout figures and results. For example, Results state MUSAP1 in Supplementary Fig. 11A. This should be NUSAP1. Also in the same figure, the Results do not reflect all

the markers shown. Same in Figure 4i; all genes in text do not match those in the figure. Is CXCL14 (in results) or CXCL12 (in Fig. 4i) increased in CAF1's?

8. Why does the lack of CAF2 in the TG group indicate that the TME of the NG group was better remodeled by CAFs than the NB group?

9. Figure 5A: Intercellular interactions look very similar – perhaps, data presented in supplementary Fig. 12a-d could be included in the main figure.

10. It is unclear why the authors extrapolated their findings from hypopharyngeal cancer and nasopharyngeal cancers which differ vastly in etiology, histopathology, and anatomic site.

Minor concerns.

1. Define TPF and TME in abstract.

2. Provide a reference for 1st sentence in Introduction. Is it 830,000 new cases of head and neck squamous cell carcinoma or 830,000 new cases of head and neck cancer? There is a difference.

3. Figures 2f through 2 k are incorrectly referenced (due to incorrect labeling?) in the results making it difficult to understand.

4. Supplementary Figure 4b: Figure legend does not correspond to figure.

5. What do the different colors represent in Supplementary Fig. 11b? A key should be provided.

6. Figure 5b through 5e: The validity of the conclusions are difficult to assess because the scale is not easily discernible, i.e. could a wider range of colors be used? Some of the colors in NB and NG are very similar.

7. Figure 6c is not mentioned in the results section although the data are described.

REVIEWER COMMENTS

Reviewer #1 (Remarks to the Author): Expert in head and neck cancers

Y. Zhang and coauthors collected HPC tumor samples before and after combination therapy and classified tumor samples into four groups (NaiveGood, NaiveBad, TreatGood and TreatBad) based on radiological diagnosis. The authors performed single cell RNA-seq to analyze HPC tumor microenvironment, and identified gene modules and cell constitutions that affect the response to combination therapy. Based on the single-cell HPC TME map, the authors deduced cell constitution from the bulk RNA-seq data of the tumors, and established a classifier model to predict the efficacy of combination therapy.

Overall, this study provides a useful resource of single-cell RNA before and after the standard clinical treatment of the rare but malignant HPC tumor, which may promote the understanding of HPC tumor progression and have the potential to improve the treatment practice. Moreover, this study shows an interesting and useful strategy for identifying prognosis indicators for tumor treatment based on single-cell and bulk RNA-seq data. I have several comments to which the authors should address:

1.1 To predict the efficacy of the combination therapy, the authors trained a SVM model using a set of HPC bulk RNA-seq data with a ROC curve of 0.86, suggesting that this model has the potential to serve as a prognostic tool. Although the HPC is a rare tumor, I encourage the authors to collect another set of HPC biopsy samples to further validate the efficiency of the SVM model.

A1.1: We thank the reviewer for this suggestion. We agree with the reviewer that it is important to collect more HPC biopsy samples for validation of the model efficiency. We collected another 6 samples from advanced HPC patients before the combined treatment. There were totally 12 samples for the final validation (detailed information in revised Supplementary Table 5). The model predicted successfully 6 out of 7 and 4 out of 5 patients for treatment-sensitive and treatment-resistant groups respectively, with the total correction rate of 10/12 () as shown in revised Fig. 6e. These results confirmed that the SVM model has the potential to serve as a prognostic tool.

1.2 Different cell types were identified in the HPC tumor samples using single-cell RNA-seq. It will be more convincing to validate these results by using immunohistochemical staining method, at least for T lymphocytes, macrophage, fibroblasts and endothelial cells.

A1.2: We thank the reviewer for this suggestion. We collected samples from RBT (responder before treatment, named as NaiveGood in our previous manuscript), NBT (non-responder before treatment, earlier named as NaiveBad), RAT (responder after treatment, earlier named as TreatGood), NAT (non-responder after treatment, earlier named as TreatBad) groups for multiplex immunohistochemical (mIHC) staining and showed results in revised Fig. 1f.

We found epithelial cells, fibroblast cells, endothelial cells and immune cells in all samples with different abundance. To be specific, we observed that there were more infiltrating immune cells in groups of RBT and RAT. Additionally, most of stained epithelial cells were identified as tumor cells

according to the scRNA-seq data analyses. The number of stained epithelial cells was significantly decreased in RAT sample compared to that in the RBT sample, while the number of stained epithelial cells was similar in NBT and NAT samples. Comparing the results of mIHC and scRNA-seq data (in revised Fig. 1f and Supplementary Fig. 3d), percentages of tumor cells and infiltrating immune cells were generally consistent, supporting our identification of various cell types through scRNA-seq.

1.3 Recently, A Asrir et. al (DOI: 10.1016/j.ccell.2022.01.002.) reported that tumor-associated high endothelial venules (TA-HEVs) co-express MECA-79+HEV sialomucins and E/P-selectins, which are involved in the tumor prognosis. Are TA-HEVs similar to the endothelial cells (Endoblood1 and Endoblood2) found in this study?

A1.3: We thank the reviewer for this comment. After careful reading of the recommended paper, we have known that high endothelial venules (HEVs) are specialized blood vessels for lymphocyte recruitment, and they are located at lymphoid organs and some human solid tumors. The latter one is called tumor-associated HEVs (TA-HEVs). Through bulk FACS and bulk RNA-seq, the group assessed different expression profiles between subtypes of HEV endothelial cells (HECs) from homeostatic lymph node (LN-HECs), inflamed lymph node (iLN-HECs), tumor mouse model (TA-HECs) and traditional MECA-79 negative ECs (TA-ECs and LN-ECs).

Using the marker genes from the paper, we tested the subtypes of Endoblood1 and Endoblood2 in our study. On the one hand, we found Endoblood1 had relatively high expressions of post-capillary venule endothelium genes and relatively low expression levels of genes for arterial and capillary endothelium (added results in revised Supplementary Fig. 11b), which was the expression pattern of all HECs including LN-HECs, iLN-HECs and TA-HECs. On the other hand, Endoblood1 showed higher level of TA-HECs specific genes shown in revised Fig. 4c.

These expression results indicated us that EndoBlood1 was likely to be in a state of TA-HECs, whose anti-tumor function mentioned in the paper was consistent with better prognosis inferred by the signature gene-set from EndoBlood1 in our HPC and public NPC cohort (in revised Supplementary Fig. 15b and Supplementary Fig. 18d). Therefore, we have updated corresponding EndoBlood1 descriptions in the revised manuscript.

1.4 The data processing and quality control are important for single-cell RNA-seq analysis. In the method, the authors described the 10x raw data process based on the number of detected genes, UMIs and doublets. How about other factor, such as cell-cycle related genes?; and what are the contaminated cells mentioned in Page 14 line 469?

A1.4: We thank the reviewer for this comment.

In the quality control filters in our scRNA-seq data, we used CellCycleScoring() function embedded in Seurat R package to check the cell-cycle effect in all major cell types, which aimed to exclude the influence of differential gene expressions of cell states in recognition of cell subtypes. As shown in Response Fig 1, we found annotated cells of three states were evenly distributed in endothelial and myeloid cells. As for fibroblast cells, T/NK and B cells, there existed subsets of cells annotated as G2M/S with high expression of cell-cycle gene MKI67. However, these subsets of cells had corresponding functions in the downstream analyses. In epithelial cells, cells annotated as G2M/S

state were scattered on the plot, indicating there were cells with high cell-cycle feature from many samples, consistent with functional gene module findings in malignant tumor cells. In summary, in our study, scRNA-seq data was not affected obviously by cell-cycle effect.

Response Fig 1

“Contaminated cells” mentioned in Page 14 line 469 means the cells we found in our study, which had dual-features from definitely two different cell types and mainly originated from the samples with relatively low cell viability in the measurement before sequencing. As an example in Response Fig 2, cluster 15 in unfiltered B cells were defined as contaminated cells. They had genes expression for both CA79A and LYZ but showed low scores for doublet estimation, and they almost originated from sample T5-2, with estimated ~70% cell viability in the measurement. So we tagged such cells as unqualified cells.

Response Fig 2

For better description and illustration of quality control filters in scRNA-seq, we draw a plot to be added in revised Supplementary Fig. 2a and updated corresponding information in Methods.

1.5 The authors should clarify how the authors used non-negative matrix factorization (NMF) to identify 61 megagenes from 11 HPC samples shown in Figure 2e.

*A1.5: We thank the reviewer for this suggestion. As shown in Fig. 2b, malignant tumor cells were derived from 13 samples. After excluding the few tumor cells from lymph and RAT (responder after treatment, earlier named as TreatGood), there left tumor cells from 11 samples for NMF analysis. The results of NMF will be random without a specific initialization, and the choice of factorization parameter (rank) is very important. As a rule of thumb, we chose rank= 4-7 for pre-test with 100 repetitions. In order to select the most appropriate rank, corresponding cophenetic correlation coefficients were recorded. The index was used to evaluate the stability of NMF results under multiple random initializations. The higher the value represented, the more stable the factorization reached under the parameter. The best rank was defined as the one before the coefficient consistently decreased¹. As shown in revised Supplementary Fig. 5a, rank was chosen as 5 for five samples of T2-1, T3-1, T5-1, T6-1, T7-2, and remaining six samples with rank = 6. Therefore, in the subsequent analysis, we conducted formal NMF based on this and obtained 61 ($5 * 5 + 6 * 6 = 61$) meta-genes.*

1.6 In the section of TME compositions and cell-cell interactions in HPC, cellular interactions between 12 conventionally defined cell types were shown in Figure 5. However, more subtypes were defined in Figure 3. Does the study of the cell-cell interactions between these subtypes provide more information on TME?

A1.6: We thank the reviewer for this comment. We tried cellular interaction analyses with detailed cell subtypes before our submission, but the results would not provide more information.

As shown in Response Fig 3, EndoBlood were divided into two subtypes for cellular interaction analysis with MalignantEpi in right panel, with the original representation in left panel. From both panels, we could observe that the overall activation of angiogenesis was lowest in RBT (responder before treatment, earlier named as NaïveGood) group and highest in NBT (non-responder before treatment, earlier named as NaïveBad). Apart from this, EndoBlood2 showed more intensive interactions compared to EndoBlood1 in the right panel, which was consistent with the former description for these two subtypes. Same conclusion could also be derived when splitting CD8+ T cells into subtypes in different developmental states, as shown in Response Fig 4. In the plot, we could reveal that immune inhibition between MalignantEpi and CD8+ T cells was strengthened as CD8+ T cell subtypes went through to exhaustion state.

Considering the above results, we merged subtypes with high similar function or subtypes in continuous states of a specific cell type, and focused on cellular interactions differences in groups.

Response Fig 3

Response Fig 4

1.7 I strongly suggest to have professional editor to proofread the English throughout the manuscript before it is accepted for publication.

A1.7: We thank the reviewer for this suggestion. We have asked professional editors to polish the English language throughout the manuscript.

Reviewer #2 (Remarks to the Author): Expert in single-cell RNA-seq, tumour microenvironment, and drug response prediction

Zhang et al. uses single-cell transcriptomics to comprehensively dissect the tumour microenvironment in hypopharyngeal carcinoma (HPC) biopsies before and after combined therapy and identified three gene modules associated with tumor progression. Bulk RNA-seq data from HPC patients is also used for the analysis of tumor gene modules to predict prognosis. Finally the authors propose a classifier to predict the relationship between cell subtype abundance in TME and the efficacy of combined treatment.

In general the manuscript is well written, fluent, clear and understandable. The authors make a great effort performing a very comprehensive dissection of TME cell subpopulations and its functional implications according to the HPC subgroups based on clinical diagnosis (NG,NB,TG,TB).

In addition, currently there are not too many single-cell datasets including cancer patients before and after combined therapy. As far as I know, there is no other published work that has presented single-cell results in HPC and with such a complete description of the cell types present in HPC microenvironment. Undoubtedly, such a single-cell dataset is of interest to the cancer research community.

Major points:

2.1 In the Discussion the authors state “In the future, the ideal application scenario is to divide treatment-naïve HPC patients into two groups after their RNA-seq data are analyzed by the classifier model. Patients tagged as “NaiveGood” will be recommended to use the “classical combined therapy” plan, and the other group predicted as “NaiveBad”, who have less potential to benefit from conventional therapeutical scheme, could choose clinical trials or other aggressive therapy to pursue beneficial hopes.”

I believe that the authors could go a step further and propose some treatments through an in silico prescription study. In other words, this study should be enriched by an analysis of in silico therapeutic approaches (see PubMedID: 34911571) for the 4 groups of HPC, especially for those subtypes that do not respond well to combination therapy (i.e. NaiveBad, Treated Bad if possible). Such a study would allow to suggest potential candidate treatments to be proposed for patients who do not respond to combined treatment. In addition, it would allow to know whether or not the greater functional heterogeneity observed in the TB group (line 142, Suppl Fig 5) corresponds to greater therapeutic heterogeneity.

In my opinion, it could be very interesting and would add more value to the work.

A2.1: We would like to express our heartfelt thanks to the reviewer for the useful suggestion. In the recommended paper (PubMedID: 34911571), the authors use two collections of drugs to estimate their effects in various tumor cell lines, and derive two collections of signature gene-sets to distinguish different subsets of tumor cells with differential drug-responsiveness and drug-sensitivity respectively.

Similarly, we extracted malignant tumor cells from RBT (responder before treatment, named as NaïveGood in our previous manuscript), NBT(non-responder before treatment, earlier named as NaïveBad) and NAT (non-responder after treatment, earlier named as TreatBad) groups, and then used signature gene-sets of assessing drug-sensitivity in tumor cells to do downstream analyses. Although we could not divide malignant cells into three groups separately based on signature scores, we uncovered a group of high-sensitivity drugs for tumor cells in NBT and NAT groups (added results in revised Fig. 6f, g and Supplementary Fig. 18). The drugs of RO-3306 and CAL-101 were identified to have highest sensitivity scores in RBT and NBT groups respectively (in revised Fig. 6f, g). RO-3306 acts as a CDK1 inhibitor to block the cell-cycle in cancer cells, and CAL-101 has been used as a medication to treat certain blood cancers^{2, 3}, both of which had the potential in clinical practice.

Meanwhile, due to the differences between tumor cell lines and patient-derived tumor cells, almost all drugs had median switch points shown in revised Supplementary Fig. 18, indicating that drugs could not affect tumor cells in a homogeneous way. Further researches need to be done to validate the efficacy of recommended drugs in PDO or PDX models.

2.2 The authors must also upload both scRNA-seq and RNA-seq HPC datasets to a publicly accessible repository to make them available to the scientific community.

A2.2: We thank the reviewer for this suggestion. The raw sequence data of both scRNA-seq and bulk RNA-seq reported in this study for HPC have been deposited in the Genome Sequence Archive-Human in the National Genomics Data Center, Beijing Institute of Genomics (China National Center for Bioinformation), Chinese Academy of Sciences, under accession number HRA003383 (accessible at <http://bigd.big.ac.cn/gsa-human>).

Minor:

2.3 Perhaps if a small table or description of the number of patients included in the paper, the type (NG, NB, TG, TB) and the type of sample (i.e. single-cell, bulk) could be included in one of the main figures. It would clarify the reading of the paper and summarise what is described by the authors from line 104 onwards.

A2.3: We thank the reviewer for this suggestion. On the one hand, we have renamed the groups throughout the revised manuscript in a more common and easier way for understanding, with explanations shown in revised Fig. 1a. On the other hand, we have updated the illustration in revised Fig. 1c with summary tables for sample collections.

2.4 Please include legend in Figure 1h

A2.4: We thank the reviewer for careful reading of our manuscript. In order to put more important

contents here and organize our revised manuscript more logically, we have put the results of immunostaining in revised Fig. 1 and removed the original Fig. 1h, which is part of the plot in revised Supplementary Fig. 1d with corresponding legend.

2.5 Figure 5a. It is almost the same Figure for NG, NB and TB. The authors should consider moving it to Supplementary. In addition, there is a typo in the header of the Circos plot, "NaiveBad" is repeated.

A2.5: We thank the reviewer for this suggestion. We have moved the original Fig. 5a to Supplementary materials, which is now shown in revised Supplementary Fig. 13a with correct headers. Moreover, we changed the way of presenting results by bar-plots to show cellular interaction differences among groups in revised Fig. 5a, with the same results from revised Supplementary Fig. 13b-e.

2.6 Figure 5b has no legend

A2.6: We thank the reviewer for careful reading of our manuscript. We have added corresponding figure legend in the revised manuscript.

2.7 Figure 6a: The columns are not sorted consistently according to the Status label (Alive and Dead). Please, sort them consistently.

A2.7: We thank the reviewer for careful reading of our manuscript. We have sorted samples consistently according to the Status label (from Alive to Dead) in revised Fig. 6a.

2.8 Typo in Figure 7: "Geneset expression".

A2.8: We thank the reviewer for careful reading of our manuscript. We have revised Fig. 7 for better illustration and double-checked the spellings to avoid mistakes.

2.9 Given that the authors have both single-cell and bulk data, they may find it interesting to use in future studies other tools that take advantage of both types of data to infer cellular composition and expression in large cohorts of bulk RNA-seq data. (e.g. PMID: 35469013)

A2.9: We thank the reviewer for this suggestion. With both of single-cell and bulk RNA data in hands, we used cell subtype signature feature matrix generated from scRNA-seq to deconvolve bulk RNA-seq samples in our HPC cohort by CIBERSORTx in the original manuscript.

We have read the recommended paper (PMID:35469013) carefully. The authors develop a new mathematical algorithm named BayesPrism for cell proportion construction with statistical marginalization utilizing matched scRNA-seq and bulk RNA-seq data. The new tool has two particular advantages. One is that the tool can accurately recover gene expression and cell proportion in heterogeneous cell types like tumor cells. The other one is that it is able to improve estimates of cell type fractions through updates of the reference matrix with a large cohort of bulk RNA-seq data.

After testing this algorithm with our data, we found that the tool was not suitable for our datasets. HPC is a relatively rare malignancy, without large datasets available for updates in the step 4 of BayesPrism, which would limit its performance. Instead of major cell types, we focused on non-

malignant cell subtypes with specific functional characteristics for cell proportion construction, which were used as a potential prognostic tool.

Considering the above reasons, we still used CIBERSORTx for deconvolution of bulk RNA-seq data in our study. Besides, we have cited the new method in the revised manuscript.

Reviewer #3 (Remarks to the Author): Expert in head and neck cancer pathology and therapy, and tumour microenvironment

The overall goal of this study was to identify the relationship between TME cell type composition and patients' response to combined therapy. Overall, this is a very interesting approach to investigating hypopharyngeal cancer with potential clinical implications. However, it is important that the following are addressed: Is assessment of the cell constitutions in TME better than the current approach for therapy selection? Additionally, it is unclear why the authors extrapolated data from hypopharyngeal cancer to nasopharyngeal cancer which differs in location, histopathology, and anatomic site. There are also several minor discrepancies between data labels in the Results section and figures.

Major concerns.

3.1 In the Kaplan-Meier curves, (2h, 2k, 2n), the number of patients at risk at each time point should be shown to facilitate data interpretation. How does survival compare between groups, i.e. EndoBlood2, CAF2 etc (log rank tests)? How do independent variables affect the survival time.

A3.1: We thank the reviewer for this suggestion.

- 1) We have added risk tables for Kaplan-Meier curves in revised Fig. 2l-n and Supplementary Fig. 6a-c. After the revision, the plots in revised Fig. 2l-n showed stratified groups divided according to median module scores, and the plots in revised Supplementary Fig. 6a-c showed stratified groups divided by mean module scores. Both results showed similar survival curves.*
- 2) We moved the original Fig. 6c & Fig. 6d of survival comparisons to the revised Supplementary Fig. 15 with corresponding legend. For these comparisons, we derived featured gene-sets specific expressed by corresponding subtypes (in revised Supplementary Table 4), utilized GSVA R package⁴ to calculate module scores for samples in our HPC cohort, and stratified them into high or low groups for survival analyses.*
- 3) In revised Fig. 3-4 and Supplementary Fig. 7-12, we characterized the tumor-promoting and tumor-suppressive biological functions of these 8 subtypes. For example, Endoblood2 that promoted tumor progression by cell migration and angiogenesis as shown in revised Fig. 4b and Supplementary Fig. 11b-d, would lead to poor prognosis (in revised Supplementary Fig. 15). Other studies^{5,6} also demonstrated specific subtypes would have effects on survival time.*

3.2 In using the cell constitutions in TME (Figure 6) to predict curative effects of combined therapy, is this better than the current approach? What is the magnitude of the difference between groups?

A3.2: We thank the reviewer for this comment. According to 2022 NCCN (national comprehensive cancer network) Clinical Practice Guidelines in Oncology, advanced HPC patients (from T2 to T4 stages) could choose to accept induction chemotherapy, partial or total laryngopharyngectomy, systemic therapy or clinical trials. In our clinical practice, TPF (taxol, cisplatin and 5-FU) plus

cetuximab (defined as the combined treatment in our study) is the preferred choice for advanced HPC patients. Most patients failed to respond to the combined treatment would lose the chance to preserve their laryngeal function as shown in revised Supplementary Fig. 16.

Here, we have developed the prediction model for these advanced HPC patients, which would enlarge the possibility to preserve their laryngeal function (revised Supplementary Fig. 16). Briefly, patients who do not respond to the combined treatment would be identified by our prediction model, and be advised for available surgery strategies or other clinical trials in time, to avoid condition deterioration or loss of full laryngeal function due to delayed surgery.

3.3 Supplementary Figure 8e: Contrary to what is reported in the results, cDC2 appears to be more of a killer subset than pDC.

A3.3: We thank the reviewer for this comment and apologize for the lack of clarity in this part. In revised Supplementary Fig. 9e (the original Supplementary Fig. 8e), we used gene-sets to compare functional differences among four DC subtypes, along with their gene expressions in revised Supplementary Fig. 9a. For cDC2, it had high scores in functions of cell differentiation and cell apoptosis, indicating it was a well-matured cell subtype gradually approaching self-death, rather than serving as a killer subtype. With high expression levels of LAMP3 gene and immune-suppression gene-set, we thus concluded that cDC2 tended to be a well-matured immunosuppressive DC subtype in the revised manuscript.

3.4 Supplementary Figure 9: The resolution of this figure is poor; the small print that presents the key of each figure is very difficult to read.

A3.4: We thank the reviewer for this comment. We have enlarged characters with small font size in revised Supplementary Fig. 10 (the original Supplementary Fig. 9). After the final acceptance of this manuscript, we will upload all figures with high-resolution.

3.5 Figure 4c: The authors report high expression of NOTCH4 in C3_EndoBlood2. This should be shown in the figure and labeled accordingly. Others mentioned in the results such as TSPAN7 are also not shown; ICAM1 is incorrectly identified as ICMA1 in the results.

A3.5: We thank the reviewer for careful reading of our manuscript and apologize for these mistakes. 1) We have added violin plots to show gene expressions of NOTCH4 and TSPAN7 in revised Fig. 4c. In the plot, EndoBlood2 had higher expression of NOTCH4 and EndoBlood1 had higher expression of TSPAN7, both of which were consistent with results mentioned in main-text.

2) We have corrected the spelling of ICAM1 in the revised manuscript.

3.6 Fig. 4f: The authors state that “Compared to the NB group, the TB group showed decreased cells with lower StalkEC scores but higher TipEC scores (Fig. 4f; Supplementary Fig. 10b, c), indicating the combined treatment could trigger antitumor effects by remodeling the vascular endothelium to decrease cell migration.” The decrease in StalkEC is only ~2%; is this significant?

A3.6: We thank the reviewer for this comment and apologize for the lack of clarity in this part. In

the revised Fig. 4d, we could see the proportions of cells with low StalkEC score but high TipEC score in Q4 decreased in both treatment conditions, from 55.84% (RBT, named as NaïveGood in our previous manuscript) to 15.79% (RAT, earlier named as TreatGood) for treatment-sensitive samples, and from 51.52% (NBT, earlier named as NaïveBad) to 42.48% (NAT, earlier named as TreatBad) for treatment-resistant samples. With these results, we inferred that the combined treatment could remodel the vascular endothelium in TME by reducing cells, which would be likely to migration and form new vessels. Corresponding descriptions have been updated in the revised manuscript for better understanding.

3.7 Please check labels throughout figures and results. For example, Results state MUSAP1 in Supplementary Fig. 11A. This should be NUSAP1. Also in the same figure, the Results do not reflect all the markers shown. Same in Figure 4i; all genes in text do not match those in the figure. Is CXCL14 (in results) or CXCL12 (in Fig. 4i) increased in CAF1's?

A3.7: We thank the reviewer for careful reading of our manuscript and apologize for these mistakes.

1) We have corrected the spelling of NUSAP1 in the revised manuscript.

2) We had added violin plots to show gene expressions of CXCL14 and IGF1 in revised Fig. 4g. Both of them had highest levels in CAF1, consistent with the results mentioned in main-text.

3.8 Why does the lack of CAF2 in the TG group indicate that the TME of the NG group was better remodeled by CAFs than the NB group?

A3.8: We thank the reviewer for this comment and apologize for the lack of clarity in this part. In the revised Fig. 4j, we observed that the percentage of CAF2 decreased from RBT to RAT (treatment-sensitive groups), whereas the proportion was slightly increased from NBT to NAT (treatment-resistant groups). Because CAF2 was likely to be mCAF, functioning in pro-tumor way by extracellular matrix organization and collagen metabolic process, we thus inferred that CAF2 in TME could be better remodeled by effective combined treatment in treatment-sensitive groups. Corresponding descriptions have been updated in the revised manuscript for better understanding.

3.9 Figure 5A: Intercellular interactions look very similar – perhaps, data presented in supplementary Fig. 12a-d could be included in the main figure.

A3.9: We thank the reviewer for this suggestion. We have moved the original Fig. 5a to Supplementary materials, which is now shown in revised Supplementary Fig. 13a. Moreover, we changed the way of presenting results by bar-plots to show cellular interaction differences among groups in revised Fig. 5a, with same results from revised Supplementary Fig. 13b-e.

3.10 It is unclear why the authors extrapolated their findings from hypopharyngeal cancer and nasopharyngeal cancers which differ vastly in etiology, histopathology, and anatomic site.

A3.10: We thank the reviewer for this comment. The main purpose to extrapolate our findings to public NPC data was to validate our methodology in other tumor types. We used public NPC data because both HPC and NPC are subtypes of head and neck squamous cell carcinoma (HNSCC). Additionally, until now public scRNA-seq data of NPC lacked various subtype collections. We think that the signature matrix of non-malignant cell subtypes are relatively conserved between tumor

types. Therefore we used the signature matrix of non-malignant cell subtypes from our HPC scRNA-seq to deconvolve the public bulk NPC cohort. As shown in revised Supplementary Fig. 17, our method unveiled that cell compositions in TME would affect tumor progression in NPC samples, as we expected. Considering there are many differences between HPC and NPC in etiology, histopathology, and anatomic site, we have revised relevant texts in the second paragraph of discussion to avoid misunderstanding.

Minor concerns.

3.11 Define TPF and TME in abstract.

A3.11: We thank the reviewer for this suggestion. We have added the definition of TPF and TME in abstract of the revised manuscript.

3.12 Provide a reference for 1st sentence in Introduction. Is it 830,000 new cases of head and neck squamous cell carcinoma or 830,000 new cases of head and neck cancer? There is a difference.

A3.12: We thank the reviewer for this comment and apologize for the lack of clarity in this part.

1) *We have cited the related paper in the revised manuscript to confirm statistics for cases of head and neck cancer occurrence and death. Moreover, the original numbers were based on 2018 global cancer statistics, and we have updated with new numbers with 2020 global cancer statistics in the revised manuscript.*

2) *Case numbers of occurrence and death were summarized for head and neck cancer, because we selected cases of lip and oral cavity, larynx, nasopharynx, oropharynx and hypopharynx from table 1 of the related reference for summary⁷. But according to the research⁸, 90% of head and neck cancer are head and neck squamous cell carcinoma (HNSCC).*

These two parts have been updated in the revised manuscript.

3.13 Figures 2f through 2 k are incorrectly referenced (due to incorrect labeling?) in the results making it difficult to understand.

A3.13: We thank the reviewer for careful reading of our manuscript and apologize for these mistakes. We have readjust the sequence numbers in revised Fig. 2, and double-checked corresponding figure legends and contents in the revised manuscript to avoid readers' confusion in this part.

3.14 Supplementary Figure 4b: Figure legend does not correspond to figure.

A3.14: We thank the reviewer for careful reading of our manuscript and apologize for these mistakes. We have updated the figure legend for Supplementary Fig. 4b.

3.15 What do the different colors represent in Supplementary Fig. 11b? A key should be provided.

A3.15: We thank the reviewer for this comment and apologize for the lack of clarity in this part. We have added the related annotations in revised Supplementary Fig. 12b (earlier shown as Supplementary Fig. 11b).

3.16 Figure 5b through 5e: The validity of the conclusions are difficult to assess because the scale is not easily discernible, i.e. could a wider range of colors be used? Some of the

colors in NB and NG are very similar.

A3.16: We thank the reviewer for this suggestion. We have chosen a different color annotation in revised Fig. 5b-e for better illustration and comparison.

3.17 Figure 6c is not mentioned in the results section although the data are described.

A3.17: We thank the reviewer for this comment. We mentioned the original Fig. 6c with incorrect reference number as Fig. 6d in Page 10 Line 317. After the revision, we have moved plots of original Fig. 6c-d to revised Supplementary Fig. 15 and summarized the results in the revised manuscript.

References:

1. Brunet JP, Tamayo P, Golub TR, Mesirov JP. Metagenes and molecular pattern discovery using matrix factorization. *Proc Natl Acad Sci U S A* **101**, 4164–4169 (2004).
2. Vassilev LT, *et al.* Selective small-molecule inhibitor reveals critical mitotic functions of human CDK1. *Proc Natl Acad Sci U S A* **103**, 10660–10665 (2006).
3. Traynor K. Idelalisib approved for three blood cancers. *Am J Health Syst Pharm* **71**, 1430 (2014).
4. Hänzelmann S, Castelo R, Guinney J. GSVA: gene set variation analysis for microarray and RNA-seq data. *BMC bioinformatics* **14**, 1–15 (2013).
5. Chen YP, *et al.* Single-cell transcriptomics reveals regulators underlying immune cell diversity and immune subtypes associated with prognosis in nasopharyngeal carcinoma. *Cell Res* **30**, 1024–1042 (2020).
6. Gong L, *et al.* Comprehensive single-cell sequencing reveals the stromal dynamics and tumor-specific characteristics in the microenvironment of nasopharyngeal carcinoma. *Nat Commun* **12**, 1540 (2021).
7. Sung H, *et al.* Global Cancer Statistics 2020: GLOBOCAN Estimates of Incidence and Mortality Worldwide for 36 Cancers in 185 Countries. *CA Cancer J Clin* **71**, 209–249 (2021).
8. Curado MP, Hashibe M. Recent changes in the epidemiology of head and neck cancer. *Curr Opin Oncol* **21**, 194–200 (2009).

REVIEWERS' COMMENTS

Reviewer #1 (Remarks to the Author):

Well done and thank you for addressing my concerns.

No Further comments are from me.

Reviewer #2 (Remarks to the Author):

The authors have addressed and responded to all requests made to them.

Reviewer #3 (Remarks to the Author):

The authors have been very responsive to the reviewers' comments. With the new abbreviations and the additional data, the manuscript is greatly improved. A few minor suggestions are included below.

Minor suggestions

Line 74 on the Introduction page 7 should this be EGFR?

Since RBT and RAT refer to the clinical samples, all RATs should have a corresponding RBT. However, in Table I, there are three patients, ages 46, 71 and 56 that do not correspond in age to any RBTs. Similar discrepancies exist in NATs. Could the authors explain the discrepancies?

Please provide risk tables in Figure 1C and Supplementary Figure 1.

The schematic in Figure 1C indicates that scRNA-seq profiles were generated from 15 patients. However, the Results and Supplementary Table 2 indicate 8 patients.

Supplementary Figure 3A: KRT19 appears to be picking up fibroblasts and B cells. Please explain.

The quantification data for Figure 1f should be shown, not just mentioned.

Supplementary Figure 5A through C: Is this the NPC cohort rather than the HPC cohort indicated in the figure legend?

In the Results section, Supplementary Figure 10A is mentioned after Figure 6. Should this actually be Supplementary Figure 7A through 7C?

There are several typographical errors in figure labels and the manuscript. For example, Figure 3b, should be 'proliferation markers'. Suggest editing the manuscript carefully.

What about Supplementary figures 9c and 9f? These are not mentioned in the text.

REVIEWER COMMENTS

Reviewer #1 (Remarks to the Author):

Well done and thank you for addressing my concerns. No Further comments are from me.

Reviewer #2 (Remarks to the Author):

The authors have addressed and responded to all requests made to them.

Reviewer #3 (Remarks to the Author):

The authors have been very responsive to the reviewers' comments. With the new abbreviations and the additional data, the manuscript is greatly improved. A few minor suggestions are included below.

Minor suggestions

3.1 Line 74 on the Introduction page ◊ should this be EGFR?

A3.1: We thank the reviewer for careful reading of our manuscript. We have checked the reference paper and corrected the spelling of gene EGFR in the revised manuscript.

3.2 Since RBT and RAT refer to the clinical samples, all RATs should have a corresponding RBT. However, in Table I, there are three patients, ages 46, 71 and 56 that do not correspond in age to any RBTs. Similar discrepancies exist in NATs. Could the authors explain the discrepancies?

A3.2: We thank the reviewer for this comment. In Supplementary Table 1, 44 samples in our HPC cohort were collected from 44 individual patients, and thus samples in RAT and NAT did not have corresponding samples in RBT and NBT groups. Because patients enrolled and dropped out of the cohort study at different stages of the combined treatment, and because RNA preparations extracted from some of the biopsy samples were degraded, the majority of samples did not have the corresponding paired ones. Additionally, in order to better demonstrate the differences in the survival time of patients in different groups, several paired samples were deleted for further analyses. We have revised relevant contents in the first paragraph of results to avoid misunderstanding.

3.3 Please provide risk tables in Figure 1C and Supplementary Figure 1. s

A3.3: We thank the reviewer for this suggestion. We have added the risk tables in revised Figure 1b and Supplementary Figure 1.

3.4 The schematic in Figure 1C indicates that scRNA-seq profiles were generated from 15 patients. However, the Results and Supplementary Table 2 indicate 8 patients.

A3.4: We thank the reviewer for this comment. In our study, we collected 15 clinical samples from 8 patients for scRNA-seq profiles, and detailed information was shown in the Supplementary Table 2. In Figure 1c, numbers in the summary table referred to the sample sizes in corresponding groups, and thus the total number was 15.

3.5 Supplementary Figure 3A: KRT19 appears to be picking up fibroblasts and B cells. Please explain.

A3.5: We thank the reviewer for careful reading of our manuscript. As shown in Response Fig 1, KRT19 was mainly expressed in epithelial cells, with quite a few expression cells in other major cell types. According to our experiences and other similar studies, the small amount of KRT19 expression in other cell types should be noise happened during scRNA-sequencing, which would not lead to misidentification of major cell types.

Response Fig 1

3.6 The quantification data for Figure 1f should be shown, not just mentioned.

A3.6: We thank the reviewer for this suggestion. We have modified related contents with the quantification data in revised manuscript for better understanding.

3.7 Supplementary Figure 5A through C: Is this the NPC cohort rather than the HPC cohort indicated in the figure legend?

A3.7: We thank the reviewer for careful reading of our manuscript. We assume you're referring here to the results in Supplementary 6a-c. The results in Supplementary 6a-c was all calculated based on our HPC cohort (samples in RBT and NBT groups) as indicated in the figure legend, serving as supplementary results to Figure 2l-n with a different stratifying method. Both of the results showed same prognostic trends of the three biological gene modules in our HPC cohort.

3.8 In the Results section, Supplementary Figure 10A is mentioned after Figure 6. Should this actually be Supplementary Figure 7A through 7C?

A3.8: We thank the reviewer for careful reading of our manuscript. Supplementary Figure 10 presented the detailed characterization of B cells based on scRNA-seq profiles, which were mentioned after describing myeloid subtypes in section 4 named "Myeloid and B cell clustering and state analysis in HPC" shown in Figure 3 and Supplementary Figure 9. Therefore, we think the order of the figures is appropriate.

3.9 There are several typographical errors in figure labels and the manuscript. For example, Figure 3b, should be 'proliferation markers'. Suggest editing the manuscript carefully.

A3.9: We thank the reviewer for careful reading of our manuscript. We have revised the spelling in Figure 3b and double-checked the typographical mistakes throughout the manuscript.

3.10 What about Supplementary figures 9c and 9f? These are not mentioned in the text.

A3.10: We thank the reviewer for careful reading of our manuscript. We have added corresponding descriptions in revised manuscript.